# Reducing Hydrological Uncertainty in Large Mountainous Basins: The Role of Isotope, Snow Cover, and Glacier Dynamics in Capturing Streamflow Seasonality

Diego Avesani<sup>1</sup>, Yi Nan<sup>2</sup>, and Fuqiang Tian<sup>2,3</sup>

Correspondence: Yi Nan (ny1209@qq.com)

Abstract. Hydrological modeling in large mountainous catchments faces challenges due to the complex interplay of snowmelt, glacier dynamics, and groundwater contributions, which introduce significant uncertainty in streamflow predictions. This study introduces a Bayesian multi-objective parameter estimation framework to reduce predictive streamflow uncertainty in large mountainous catchments by integrating streamflow likelihood with three auxiliary likelihoods, analyzed individually: snow cover area (SCA), glacier mass balance (GMB), and isotopic composition (I). The well-established Generalized Likelihood Uncertainty Estimation (GLUE) method is employed to investigate trade-offs among these likelihoods, providing a detailed assessment of their distinct and combined contributions to hydrological model performance across various flow regimes. The semi-distributed Representative Elementary Watershed-Tracer aided version (THREW-T) hydrological model applied in this work captures both rapid surface dynamics and slow-response subsurface processes, offering a comprehensive representation of streamflow variability.

Results indicate that isotopic likelihood plays a critical role in reducing low-flow uncertainty by effectively constraining subsurface flow and groundwater-surface water interactions, particularly during winter and early spring when these processes dominate. Conversely, while SCA and GMB likelihoods demonstrate some effectiveness in capturing rapid processes such as snowmelt and glacier melt, their influence is most pronounced during the melting season, with limited impact on reducing overall streamflow uncertainty. This seasonality is reflected in sharpness values, which measure how much uncertainty is reduced, with isotopic likelihood achieving the highest peak of 0.34 in late winter, whereas SCA and GMB reach maximum sharpness values of 0.19 and 0.16, respectively, during the melting season. Pareto plots further reveal the synergies and trade-offs associated with each likelihood, underscoring the importance of adopting a multi-objective calibration approach that accounts for seasonal variations in hydrological processes. In addition, the results highlight the critical role of seasonality in shaping the effectiveness of auxiliary likelihoods, emphasizing their potential to improve predictive accuracy and reduce uncertainty in hydrological models.

Copyright statement. TEXT

<sup>&</sup>lt;sup>1</sup>Department of Civil, Environmental and Mechanical Engineering, University of Trento, 38123 Trento, Italy

<sup>&</sup>lt;sup>2</sup>State Key Laboratory of Hydroscience and Engineering, Tsinghua University, Beijing 100084, China

<sup>&</sup>lt;sup>3</sup>Department of Hydraulic Engineering, Tsinghua University, Beijing 100084, China

#### 1 Introduction

30

40

Accurate hydrological modeling in large mountainous catchments remains particularly challenging due to the inherent complexity of these systems (Gupta et al., 2008). The interplay of multiple water sources, such as snowmelt, glacier dynamics, and groundwater, combined with substantial spatio-temporal variability in streamflow generation, often results in equifinality and significant uncertainty in predictions (e.g., Asong et al., 2020; Shuai et al., 2022; Dalla Torre et al., 2024). These complexities call for advanced modeling approaches capable of improving our understanding of streamflow variability and supporting effective water resource management (Panchanathan et al., 2024).

Recent advancements in hydrological modeling have addressed these demands by focusing on the integration of auxiliary variables, such as snow cover area, glacier mass balance, and environmental tracers (e.g., stable oxygen isotopes,  $\delta^{18}O$ ), to improve model calibration and reduce parameter uncertainty (Di Marco et al., 2021; Nan et al., 2021b; Mohammadi et al., 2023). These variables provide critical insights into cryospheric and subsurface processes, enabling models to better capture hydrological responses that drive streamflow variability during periods of low flow (Panchanathan et al., 2024). Incorporating such data improves the representation of specific model components and guides the evaluation of the model, ultimately enhancing reliability and reducing equifinality (Birkel et al., 2014; Tetzlaff et al., 2014). Tracer-aided modeling has proven particularly effective in disentangling hydrological processes and identifying critical contributions from snowmelt and groundwater under varying conditions (Nan et al., 2021b). Bayesian approaches have also been applied to explicitly address equifinality and uncertainty in hydrological modeling in various mountain basins (e.g., Yang et al., 2007; Andraos, 2024).

Nonetheless, several challenges remain. Few studies have systematically compared the relative effectiveness of auxiliary datasets - such as snow cover area, glacier mass balance, and isotopic tracers - in reducing model uncertainty and equifinality across different flow regimes (Finger et al., 2011; Xu et al., 2012; Nan and Tian, 2024). While some studies have explored the role of individual datasets, such as isotopic tracers (Nan and Tian, 2024) or glacier mass balance (Finger et al., 2011), a unified comparison of their respective contributions within a single modeling framework remains absent. This is particularly true for low-flow conditions, which are often dominated by slow-response processes such as groundwater contributions and subsurface flow dynamics (Betterle and Bellin, 2024). Moreover, the potential for these datasets to improve the representation of hydrological processes under varying seasonal conditions remains largely unexplored. Similarly, while previous work has explored the Contributions of Runoff Components (CRC) to total streamflow (e.g., subsurface flow, rainfall runoff, snowmelt, and glacier melt) (Stahl et al., 2008), a comprehensive understanding of how these components interact to influence streamflow dynamics under different conditions remains insufficiently constrained by multi-source datasets. Current Bayesian frameworks, while powerful, often fail to fully leverage the complementary strengths of auxiliary datasets, particularly in large mountainous catchments where complex cryospheric and subsurface interactions drive streamflow dynamics (Zhang et al., 2018; Chang et al., 2024).

This study addresses these gaps by systematically evaluating the role of snow cover area, glacier mass balance, and isotopic tracers in reducing model uncertainty and equifinality within a fully Bayesian framework. Focusing on the Yarlung Tsangpo River Basin—a large mountainous catchment where streamflow variability arises from snowmelt, glacier dynamics,

and groundwater contributions—we investigate how these auxiliary datasets can complement each other in constraining hydrological models across different flow regimes. Special emphasis is placed on low-flow periods, during which isotopic data provide particularly strong constraints on subsurface flow and groundwater–surface water interactions (Rodgers et al., 2005). By adopting a multi-source calibration approach, we explore trade-offs in model performance and quantify how each dataset influences the contributions of snowmelt, glacier melt, rainfall runoff, and subsurface flow. By shedding light on streamflow generation processes, particularly during low-flow periods, these findings may offer a first step toward more integrated and nuanced water management strategies in complex mountainous regions facing increasing drought risk (Wu et al., 2023).

To address these objectives, the paper is organized as follows: the adopted tracer-aided hydrological model, the study area, and the Bayesian framework are described in Section 2. Section 3 presents the results, including parameter distributions, uncertainty analysis, and flow regime-specific improvements. Section 4 discusses the implications of the findings, while Section 5 provides concluding remarks and future research directions.

#### 2 Materials and methods

## 2.1 Study area and data

The Yarlung Tsangpo River (YTR) basin was selected as the focus area of this study (Figure 2). The YTR basin is the upstream part of the Brahamaputra River basin, located on the southern Tibetan Plateau (TP). The YTR basin, as one of the longest rivers originating from the TP, extends in the range of 27-32°N and 82-97°E with an elevation extent of 2900-6900 m above sea level. The outlet hydrological station of the YTR basin is the Nuxia station, with a drainage area of approximately  $2 \times 10^5 \, \mathrm{km}^2$ . There are four hydrological stations along the mainstream of YTR: Nuxia, Yangcun, Nugesha and Lazi, from downstream to upstream (Table 1). During 1990-2015, the mean annual precipitation in the YTR basin is around 490 mm, which is dominated by the South Asian monsoon in the Indian Ocean hydrosphere-atmosphere system resulting an obvious wet season during June to September. The mean annual temperature is -1.5°C, leading to widely distributed snow and glacier, covering around 16.3% and 1.5% of the basin.

Datasets of meteorological input, topography, underlying surface, streamflow and isotope were collected to establish the model. The 30 m resolution digital elevation model (DEM) were downloaded from the Geospatial Data Cloud (https://www.gscloud.cn) for simulation unit dividing. Daily precipitation and temperature were extracted from the 0.1° China Meteorological Forcing Dataset (Yang and He, 2019), which was produced by merging multiple satellite datasets with the national meteorological station data. The daily potential evapotranspiration were obtained from the 1.0° reanalysis dataset ERA5\_Land (Muñoz-Sabater et al., 2021). For the underlying conditions, the MODIS leaf area index (LAI) product MOD15A2H (Myneni et al., 2015) and the normalized difference vegetation index (NDVI) product MOD13A3 (Didan, 2015) were used to represent the vegetation conditions and determine the ratio of vegetation covered area, and the Marmonized World Soil Database (He, 2019) was used to estimate the soil property parameters not obtained by model calibration (including saturated hydraulic conductivity, soil porosity, soil pore distribution index, field capacity, and air entry value). For the cryospheric elements, the second glacier inventory dataset of China (Liu, 2012) was adopted to determine the boundary of regions where glacier simulation should

be performed. The daily Tibetan Plateau Snow Cover Extent (TPSCE) product (Chen et al., 2018) during 2001-2015 and the 0.5° yearly glacier elevation change dataset developed by Hugonnet et al. (2021) during 2001-2010 were used to validate the simulated snow cover area (SCA) and glacier mass balance (GMB). Daily streamflow observation data at the Nuxia, Yangcun, and Nugesha stations were collected to evaluate the performance of the hydrological simulations. However, due to Chinese national regulations, streamflow data for the Yarlung Tsangpo River—a transboundary river system—are considered sensitive and classified as confidential. As such, these data cannot be publicly disclosed or shared in this publication. This restriction reflects broader geopolitical concerns, as highlighted by Lin et al. (2023), who emphasize the particular sensitivity of hydrological data in transboundary basins and regions subject to resource and geopolitical tensions. Considering the availability period of the supporting datasets, the simulation period was set from 2001.01.01 to 2015.12.31, aligned with the time span of the meteorological and vegetation input data.

Grab samples of stream and precipitation water were collected in 2005 at four stations to analyze the isotope composition ( $\delta^{18}O$ ) to validate the tracer simulation (Table 1). The outputs of the Scripps Global Spectral Model with an isotope incorporated (isoGSM, Yoshimura et al. (2008)) with 1.875° resolution were extracted to represent the spatiotemporal variation of the isotope composition in precipitation. Our previous evaluation on isoGSM (Nan et al., 2021a) indicated that it can effectively capture the seasonal variation in precipitation  $\delta^{18}O$ , but exhibited a systematic overestimation bias in the study region and performed relatively poorly in accurately capturing the isotope signature of specific events (Supplementary Figures S1 and S2). The bias corrected isoGSM product produced by Nan et al. (2022) was adopted as the input data, in which the bias of isoGSM was adjusted based on a linear regression with altitude. Rainfall and snowfall were assumed to have the same isotope composition as the precipitation  $\delta^{18}O$  in isoGSM. The glacier meltwater  $\delta^{18}O$  is calculated using the offset-parameter method, in which the glacier meltwater  $\delta^{18}O$  is assumed to be temporally constant and 5% lower than the weighted average of local precipitation  $\delta^{18}O$ . The value of the offset parameter was estimated from the data collected by Boral and Sen (2020).

**Table 1.** Data and sample information at four hydrological stations adopted

| Station | Coordinate       | Elevation (m) | Streamflow | Isotope          |                   |                                      |                   |                                     |
|---------|------------------|---------------|------------|------------------|-------------------|--------------------------------------|-------------------|-------------------------------------|
|         |                  |               | Period     | Period (in 2005) | Precipitation     |                                      | Stream water      |                                     |
|         |                  |               |            |                  | number of samples | $\delta^{18}{\rm O}\left(\%e\right)$ | number of samples | $\delta^{18}{\rm O}\left(\%\right)$ |
| Nuxia   | 94.65°E, 29.47°N | 3691          | 2001–2015  | 14 Mar-23 Oct    | 86                | -10.33                               | 34                | -15.74                              |
| Yangcun | 91.82°E, 29.27°N | 4541          | 2001-2010  | 17 Mar-5 Oct     | 59                | -13.14                               | 30                | -16.57                              |
| Nugesha | 89.71°E, 29.32°N | 4715          | 2001-2010  | 14 May-22 Oct    | 45                | -14.29                               | 25                | -17.84                              |
| Lazi    | 87.58°E, 29.12°N | 4889          | /          | 6 Jun-22 Sep     | 42                | -17.41                               | 22                | -16.52                              |

#### 2.2 The tracer-aided hydrological model




A semi-distributed tracer-aided cryospheric-hydrological model, Tsinghua Representative Elementary Watershed-Tracer aided version (THREW-T) developed by Tian et al. (2006) and Nan et al. (2021b) was adopted to simulate the hydrological,

cryospheric and isotopic processes in the YTR basin (Figure 1). The THREW-T model uses the representative watershed method (REW) for spatial discretization, which divides the whole catchment into REWs based on DEM data. Two vertical layers including eight subzones (i.e., surface layer including vegetation zone, bare soil, sub-stream network zone, snow-covered zone, glacier-covered zone and main channel reach zone; subsurface layer including unsaturated zone and saturated zone) are defined for each REW-based on the underlying surface type. The YTR basin was divided into 297 REWs with average area of 694 km<sup>2</sup> in this study. The areal averages of the gridded datasets were calculated for each REW, which were used as the input for simulation. More detailed descriptions of REW method could be found in Reggiani et al. (1999) and Tian et al. (2006).

Figure 1. Schematic representation of the THREW-T model




The cryospheric module was incorporated into the model to simulate the evolutions of snowpack and glacier. The total precipitation was partitioned into liquid (rainfall) and solid precipitation (snowfall), according to a temperature threshold set as 0 °C. For the simulation of snowpack, the snow water equivalent of each REW was updated based on the snowfall and the snowmelt, which was calculated using the degree-day factor method. The snow cover area (SCA) was determined by the snow cover depletion curve (Fassnacht et al., 2016) and then compared with the satellite observation data. The snow sublimation was not simulated in the model, because previous studies estimated that the sublimation losses in the study region only accounted for 2 3% of the annual snowfall, as the results of the wet climate condition (Khanal et al., 2021; Sun et al., 2024; Lutz et al., 2016). For the simulation of glacier, each REW was further divided into several elevation bands with an interval of 200m, to represent the variation in temperature and precipitation along the altitudinal profile. The glacier within the intersection of each REW and elevation band was regarded as the representative unit for glacier simulation. The processes related to glacier evolution in the model included the snow accumulation and snowmelt over glaciers, the turnover of snow to ice, and the ice melt. The ice melt was also calculated using the temperature index method but with a different degree-day factor from snowmelt. The volume of the glacier was updated based on the mass balance equation and was transferred to the glacier cover area based on a scale equation (Grinsted, 2013). The glacier melt was assumed to generate streamflow directly through the

surface pathway, considering the low permeability of glacier surface. The output of the glacier simulation included the glacier mass balance (GMB) and the glacier cover area, and the simulated GMB would be compared with the measurement data. More details of the cryospheric module can be found in Nan et al. (2021b) and Cui et al. (2023).



The tracer module was incorporated into the model to simulate the isotope composition in multiple water bodies, which characterized the isotopic variations during water mixture and phase change processes. The isotope fractionation during water evaporation and snowmelt processes was simulated by the Rayleigh equation (Hindshaw et al., 2011). The isotope compositions in each simulation unit were calculated based on the complete mixing assumption, meaning that the tracer concentration homogeneity within a unit was achieved during a simulation time step (Nan et al., 2023). Forced by the precipitation isotope input, the model can simulate the isotope composition of all the water bodies, including river water, groundwater and snowpack, and the simulated isotope composition of river water would be compared with the observation data. More details of the tracer module are provided in Nan et al. (2021b).

The Contributions of Runoff Components (CRC) were analyzed to better understand the influence of multiple datasets on hydrological simulations. Two definitions are commonly used to quantify CRC (He et al., 2021). One is based on water sources, describing where the water originates; under this definition, the three components are rainfall, snowmelt and glacier melt. The other is based on the runoff generation pathway, describing how water produces runoff; here, the two components are surface and subsurface runoff. The THREW-T model quantified the runoff components based on the definition that combines water sources and runoff generation pathways. Specifically, the runoff was first divided into surface runoff and subsurface runoff based on the runoff generation pathway. The surface runoff was further divided into three components induced by different water sources: rainfall, snowmelt, and glacier melt. Consequently, the total runoff was divided into four components: subsurface runoff, rainfall surface runoff, snowmelt surface runoff, and glacier melt surface runoff.

Figure 2. The location and topography of (a) the Tibetan Plateau and (b) the Yarlung Tsangpo River basin

**Table 2.** Parameter table with descriptions, ranges, and units.

| Symbol                      | Range  | Units                              | Description                                                 |
|-----------------------------|--------|------------------------------------|-------------------------------------------------------------|
| nt                          | 0-0.2  | _                                  | Manning roughness coefficient for hillslope                 |
| WM                          | 0–10   | m                                  | Tension water storage capacity used to calculate the sat-   |
|                             |        |                                    | uration area                                                |
| В                           | 0–1    | _                                  | Shape coefficient used to calculate the saturation area     |
| Gatr                        | 0–10   | _                                  | Coefficient representing spatial heterogeneity of ex-       |
|                             |        |                                    | change term between t-zone and r-zone                       |
| KKA                         | 0–6    | _                                  | Exponential coefficient to calculate the subsurface         |
|                             |        |                                    | runoff outflow rate                                         |
| KKD                         | 0-0.5  | _                                  | Linear coefficient to calculate the subsurface runoff out-  |
|                             |        |                                    | flow rate                                                   |
| DDFs                        | 0–10   | mm°C <sup>-1</sup> d <sup>-1</sup> | Degree-day factor for snowmelt                              |
| $\mathrm{DDF}_{\mathrm{G}}$ | 0–10   | mm°C <sup>-1</sup> d <sup>-1</sup> | Degree-day factor for glacier melt                          |
| LL                          | 0–1    | _                                  | Coefficient to transfer snow water equivalent to snow       |
|                             |        |                                    | cover area using snow depletion curve                       |
| $T_0$                       | -5 – 5 | °C                                 | Temperature threshold above which snow and glacier          |
|                             |        |                                    | melting occurs                                              |
| α                           | 0–1    |                                    | Coefficient in the Muskingum method for runoff con-         |
|                             |        |                                    | centration calculation                                      |
| β                           | 0–1    |                                    | The proportion to the $\alpha$ coefficient in the Muskingum |
|                             |        |                                    | method for runoff concentration calculation                 |

## 2.3 Multi-objective Parameter Estimation






The uncertainty estimation of model parameters was performed using the Generalized Likelihood Uncertainty Estimation (GLUE) methodology (Beven, 2006). GLUE employs Monte Carlo simulations to generate a large ensemble of model realizations, where each realization corresponds to a specific parameter set associated with a likelihood measure. Unlike traditional optimization methods that focus on identifying a single *best* parameter set, GLUE emphasizes equifinality by retaining an ensemble of acceptable parameterizations (Efstratiadis and Koutsoyiannis, 2010; Brazier et al., 2000), thus acknowledging that multiple parameter sets can produce similarly good simulations, which is particularly important when modeling complex hydrological systems where uncertainties in processes and inputs can lead to varied but equally plausible outcomes (Di Marco et al., 2021).

The selection of likelihood measures and thresholds to distinguish behavioral from non-behavioral simulations is inherently subjective and problem-dependent (Blasone et al., 2008; Jin et al., 2010). In this study, the parameter space was sampled using Latin Hypercube Sampling (LHS) (McKay et al., 1979), assuming a uniform distribution for all parameters listed in Table 1. In the absence of prior information, all parameter sets were initially considered equally probable, ensuring non-informative priors (e.g., Gan et al., 2018; Teweldebrhan et al., 2018). The impact of this uniformity assumption on posterior results was evaluated through sensitivity analyses.

A total of 25,000 parameter sets were generated and evaluated using a likelihood measure to quantify model performance. Behavioral simulations were identified based on a predefined threshold, the value of which is provided in the results section. Non-behavioral simulations were assigned a likelihood of zero, while the likelihood values of retained simulations were rescaled to sum to one, forming a posterior probability density function for the model parameters.

Predictive uncertainty of outputs, such as streamflow, was assessed by ranking behavioral simulations according to their rescaled likelihoods. The empirical cumulative distribution, weighted by these likelihoods, was used to define uncertainty bounds by excluding the lower and upper 5th percentiles (Teweldebrhan et al., 2018; Franks et al., 1998).

The Nash-Sutcliffe Efficiency Index (NSE) (Nash and Sutcliffe, 1970) was selected as the likelihood measure for stream-flow, snow-covered area (SCA), and isotopic composition (I) (Lamontagne and Barber, 2020; Araya et al., 2023), while the Volumetric Deviation Efficiency (VE) (He et al., 2018) was adopted for glacier mass balance (GMB). These two metrics were chosen to reflect both dynamic performance and cumulative accuracy across key hydrological variables.

The NSE was used as the likelihood measure for streamflow, snow-covered area, and isotopic composition. Its formulation is provided for completeness:

$$NSE_X = 1 - \frac{\sum_{t=1}^{N} (X_{sim}(t) - X_{obs}(t))^2}{\sum_{t=1}^{N} (X_{obs}(t) - X_{obs,mean})^2},$$
(1)

where X represents the variable of interest,  $X_{\text{sim}}(t)$  and  $X_{\text{obs}}(t)$  are the simulated and observed values at time step t,  $X_{\text{obs,mean}}$  is the mean of the observed values, and N is the number of time steps.

For glacier mass balance (GMB), the Volumetric Deviation Efficiency (VE) was deemed more appropriate as it directly evaluates the accuracy of the simulated mean relative to the observed mean, aligning better with the cumulative nature of GMB:

$$VE_{GMB} = 1 - \frac{GMB_{\text{mean,sim}} - GMB_{\text{mean,obs}}}{GMB_{\text{mean,obs}}},$$
(2)

where  $GMB_{\text{mean,sim}}$  and  $GMB_{\text{mean,obs}}$  are the simulated and observed mean glacier mass balances, respectively.

The multi-objective parameter estimation followed an informal Bayesian framework. The streamflow likelihood,  $LH(Q|p_i)$ , was first used to constrain the model parameters, forming the prior likelihood distribution. Auxiliary variables (X) were then incorporated to produce a posterior likelihood distribution (cLH), defined as:

$$cLH(p_i|Q,X) = \frac{1}{C} \cdot LH(Q|p_i) \cdot LH(X|p_i), \tag{3}$$

where  $p_i$  represents a parameter set,  $LH(Q|p_i)$  and  $LH(X|p_i)$  are the likelihoods for streamflow and auxiliary variables, respectively, and C is a normalization constant ensuring:

$$\int cLH(p_i|Q,X)dp_i = 1.$$
(4)

In the absence of explicit guidelines for auxiliary datasets, except for streamflow, a threshold of NSE > 0 and VE > 0, commonly used as minimal performance criteria, was systematically applied to all target variables, including streamflow (Q), snow-covered area (SCA), glacier mass balance (GMB), and isotopic composition (I). The use of NSE > 0 for streamflow ensures consistency across all metrics, even though stricter thresholds are typically recommended to ensure the reliability of streamflow simulations (Moriasi et al., 2007). Furthermore, following Di Marco et al. (2021); Ma et al. (2024), the 75th percentile was chosen as the cutoff for both the prior and posterior distributions to select parameter sets, ensuring a consistent and robust identification of the most likely parameters while balancing model accuracy and diversity.

#### 205 2.4 Metrics for Quantifying Uncertainty


To assess the added value of multi-objective model conditioning compared to single-objective approaches based solely on streamflow observations, we utilized two uncertainty metrics: the first, known as the containing ratio (CR), evaluates the ability of the simulated prediction intervals to capture the observed values and reads as follows (e.g., Teweldebrhan et al., 2018; Jin et al., 2010):

$$CR = \frac{1}{N} \sum_{t=1}^{N} \Gamma(Q_{\text{obs}}(t); Q_{\text{sim}, 0.05}(t), Q_{\text{sim}, 0.95}(t)),$$
 (5)

where  $Q_{\text{sim}0.05}(t)$  and  $Q_{\text{sim}0.95}(t)$  indicate the lower and upper bounds of the simulated 90% streamflow prediction interval, respectively, while  $\Gamma$  returns a value of 1 if the observation falls within the prediction interval and 0 otherwise. A higher CR value indicates that the prediction intervals are better at capturing observed values, reflecting improved reliability of the

model outputs. Conversely, a lower CR suggests that the prediction intervals fail to encompass the observed data as effectively, indicating potential deficiencies in the model's calibration or input data.

The second metric, the so-called sharpness (SH), is a measure that quantifies the reduction in prediction uncertainty achieved through the integration of additional information and reads as follows:

$$SH = 1 - \frac{cLH(p_i \mid Q, X)}{LH(Q \mid p_i)}.$$
(6)

A higher SH value signifies that the prediction intervals are narrower, implying reduced uncertainty in the model's predictions and a more precise representation of the streamflow dynamics. On the other hand, a lower SH value suggests broader prediction intervals, indicative of higher uncertainty or less precise modeling.

It is worth noticing that in an ideal scenario, a perfectly constrained model would achieve CR and SH values close to 1. In practice, this would imply that the prediction intervals consistently capture observed values (CR=1) and that the model uncertainty diminishes to the point where the simulated output closely aligns with the observations, indicating that there is no uncertainty in the predictions.

#### 3 Results






#### 3.1 Behavioral simulations

For each run of the overall Monte Carlo ensemble, we computed likelihood values for streamflow ( $NSE_Q$ ) and for the additional performance metrics: Snow Cover Area likelihood ( $NSE_{SCA}$ ), Glacial Mass Balance likelihood ( $VE_{GMB}$ ), and Isotope likelihood ( $NSE_I$ ). The bi-objective relationships between  $NSE_Q$  and each of these metrics are illustrated in Figure 3, where each panel shows the distribution of the full ensemble of simulations. Specifically, the red markers indicate simulations on the Pareto front, defined as the subset of ensemble members that are not dominated with respect to the two metrics shown. In other words, a simulation is considered non-dominated (i.e., Pareto-optimal) if no other simulation in the ensemble performs at least as well in both objectives and strictly better in at least one (e.g., Yapo et al., 1998; Efstratiadis and Koutsoyiannis, 2010). These points represent thus optimal trade-offs between the two objectives, as improving one necessarily implies a deterioration in the other. The Pareto front was computed over the full ensemble and independently of any behavioral classification, so red markers should not be interpreted as behavioral simulations. The blue lines in each panel indicate the thresholds used to define behavioral solutions and are included solely for visual reference. The relatively small number of Pareto-optimal simulations reflects the selective nature of such trade-offs, as most parameterizations are dominated in at least one objective. This is consistent with findings by Di Marco et al. (2021), who showed that Pareto-optimal solutions typically represent only a small subset of behavioral ones. The remaining gray points correspond to dominated simulations and delineate the broader trade-off landscape, offering insight into the variability of model performance across the ensemble.

The Snow Cover Area likelihood ( $NSE_{SCA}$ ) exhibits a strong positive relationship with streamflow likelihood ( $NSE_Q$ ). As shown in Figure 3.a, the Pareto front points (red markers) are concentrated in the upper-right quadrant of the plot, indicating that high streamflow likelihood values can coexist with high  $NSE_{SCA}$  values. This suggests strong compatibility between

these two objectives, meaning that improving streamflow performance does not inherently result in a reduction in  $NSE_{SCA}$ . The dominated solutions (gray points) show a wider spread across the plot, including regions where both  $NSE_Q$  and  $NSE_{SCA}$  values are low. This indicates variability in model performance when considering different parameter sets. The clustering of Pareto-optimal solutions in the high-likelihood region reflects the shared role of snow processes in regulating both streamflow and snow cover dynamics suggest that it is possible to improve  $NSE_{SCA}$  without significant trade-offs when calibrating the model to optimize streamflow performance.




The Glacial Mass Balance likelihood ( $VE_{GMB}$ ) shows a slightly different behavior, as illustrated in Figure 3.b. Although high streamflow likelihood values are still associated with moderate to high  $VE_{GMB}$  values on the Pareto front, the vertical spread of the red markers is more pronounced. This indicates a weaker synergy between these two metrics compared to  $NSE_{SCA}$ . While some Pareto-optimal solutions achieve high likelihoods for both  $NSE_Q$  and  $VE_{GMB}$ , others show intermediate  $VE_{GMB}$  values despite high  $NSE_Q$  performance. This pattern suggests the presence of moderate trade-offs, where accurately capturing glacial mass dynamics might be compromised to achieve better streamflow performance.

The Isotope likelihood (NSE<sub>I</sub>) exhibits the most significant trade-offs among the three metrics, as illustrated in Figure 3.c. The Pareto front (red markers) is notably dispersed, with even the highest-performing solutions for NSE<sub>Q</sub> rarely exceeding an NSE<sub>I</sub> value of 0.4. This indicates a high degree of independence and conflict between these two metrics. The complexity of this relationship is further emphasized by the dominated solutions (gray points), where many configurations achieve high NSE<sub>Q</sub> values but fail to yield satisfactory NSE<sub>I</sub> values.

Figure 3. Pareto fronts (red markers) of streamflow likelihood (NSE<sub>Q</sub>) and likelihood metrics for (a) Snow Cover Area likelihood (NSE<sub>SCA</sub>), (b) Glacial Mass Balance likelihood (VE<sub>GMB</sub>), and (c) Isotope likelihood (NSE<sub>I</sub>). The thin blue lines represent the performance thresholds defined for the multi-objective behavioral selection: NSE<sub>Q</sub> = 0, NSE<sub>SCA</sub> = 0, NSE<sub>I</sub> = 0, and NSE<sub>GMB</sub> = 0. The dominated solutions are shown as gray points.

#### 3.2 Prior and posterior parameter distributions

Figure 4 shows the prior (black lines) and posterior parameter distributions, conditioned on the likelihoods of Snow-Cover Area (SCA, orange dashed lines), Glacier Mass Balance (GMB, light blue dash-dotted lines), and isotope concentrations (IS, green dotted lines). All distributions are derived from the Monte Carlo ensemble, but only simulations with a Nash–Sutcliffe Efficiency for streamflow ( $NSE_Q > 0$ ) are retained. This threshold ensures that simulations outperform the climatological mean, thereby meeting a minimum criterion for behavioral plausibility. These behaviorally plausible simulations define the prior distribution, which is then formally updated within a Bayesian framework using the likelihoods associated with the additional observational datasets (i.e., SCA, GMB, and I). Visual inspection of the posterior distributions indicates that, in general, each dataset provides meaningful information to constrain parameters related to the physical processes it represents.

For example, the parameters  $DDF_S$  and LL (Figures 4.g and 4.i), which control snow cover area transfer and snowmelt processes, show a stronger response when conditioned on the likelihood of SCA, highlighting their direct influence on snow dynamics. While  $DDF_S$  regulates the magnitude of snowmelt, LL primarily affects the spatial extent and persistence of snow cover. As such, its influence is more pronounced in shaping the spatial and temporal patterns of snow accumulation and melt, rather than the total amount of snowmelt contributing to runoff. This explains why the posterior of LL is well constrained under SCA conditioning, but does not manifest as clearly in metrics focused on water yield, such as snowmelt fraction. Similarly, the parameter  $DDF_G$  (Figure 4.h), which governs glacier melt processes, exhibits tighter posterior constraints when conditioned on the GMB likelihood, reflecting its strong connection to ice melt dynamics. Interestingly, the parameter  $DDF_S$  shows a contrasting response under the GMB likelihood, with the posterior distribution shifting in the opposite direction compared to the SCA posterior distribution.

A similar observation can be made for the isotopic likelihood, which effectively constrains parameters related to subsurface flow and runoff partitioning. For example, the parameter KKA (Figure 4.e), which defines the subsurface runoff outflow rate, shows noticeable convergence when conditioned on isotope data. Although both KKA and KKD influence subsurface runoff outflow, only KKA shows a marked posterior convergence. This is likely due to its exponential formulation, which increases its sensitivity to the calibration targets, whereas KKD, as a linear coefficient, exerts a more gradual influence that is harder to isolate and constrain. Other parameters, such as the tension water storage capacity WM (Figure 4.b) and the shape coefficient B (Figure 4.c), which influence the calculation of the saturation area, also exhibit tighter posterior distributions, underscoring the capacity of isotope data to inform processes related to water storage and release in the subsurface. Furthermore, the runoff concentration coefficients  $\alpha$  and  $\beta$  (Figures 4.k and 4.l) are better estimated with the inclusion of isotopic data with respect to the likelihoods of SCA and GMB.

An interesting case is the temperature threshold parameter  $T_0$  (Figure 4.j), which defines the threshold above which snow and glacier melting occur. The SCA likelihood has the strongest influence on the posterior distribution of  $T_0$ . However, both the GMB and the isotopic likelihoods can narrow the posterior distribution of  $T_0$ , albeit to a lesser extent, indicating that the glacier mass balance and the isotopic data provide complementary constraints on this parameter.

In contrast, the posterior distribution of the parameter Gatr shows minimal variation compared to the previous (Figure 4. d), aligning with expectations, as Gatr reflects spatial heterogeneity, which reduces its sensitivity to individual physical processes. It is also worth noting that for the parameter nt, not only does none of the data sets (SCA, GMB, or I) significantly constrain the posterior distribution compared to the prior, but the isotopic likelihood appears counterproductive in this case, as it increases the uncertainty by broadening the posterior distribution and reducing its peak.

#### 3.3 Streamflow simulation uncertainty range






The prior and posterior likelihood distributions, as described in Section 2, were used to estimate the 5th–95th percentile prediction uncertainty ranges for daily streamflow simulations. Figure 5 illustrates these predictive uncertainty ranges in comparison to observed streamflow data recorded at the Nuxia gauging station. Due to dissemination restrictions imposed by the data provider, streamflow values are presented in normalized form throughout the figure. Specifically, a linear normalization is applied to the time series panels to enable relative comparison of flow magnitude over time, while a logarithmic normalization is used in the flow duration curves, as is standard in FDC representation, to facilitate comparison across the full range of discharges. The prior uncertainty, represented by dark grey bands, corresponds to the hydrological model conditioned solely on observed streamflow. By contrast, the posterior uncertainty ranges, shown as lighter bands, result from the integration of additional datasets; snow cover area (SCA), glacier mass balance (GMB), and isotopic data. Overall, the uncertainty bands are effective in capturing the observed streamflow values. This is confirmed by the containing ratio (CR) metric, which indicates that the prior distribution encloses approximately 96% of the observations (CR = 0.959). Posterior distributions derived from isotopic likelihoods show a slightly lower coverage (CR = 0.921), while those incorporating SCA and GMB yield CR values of 0.947 and 0.960, respectively. These results suggest that, although SCA and GMB maintain similar levels of coverage compared to the prior, they do not lead to a substantial reduction in predictive reliability. Conversely, the posterior conditioned on isotopic data demonstrates a modest decrease in coverage, indicating a more selective constraint on the model's predictive range.

Visual inspection of Figure 5 indicates no reductions in uncertainty bands for higher streamflow values across all scenarios. On the contrary, the most pronounced contraction of predictive uncertainty occurs during low-flow periods when the model is conditioned with isotopic data (Figure 5.e), whereas conditioning with SCA and GMB does not produce comparable reductions, Figures 5.a and 5.c respectively. Besides, Flow Duration Curves (FDCs), presented in the right panels of Figure 5, provide further insights into the impact of these datasets across different flow regimes. For SCA and GMB Figure 5.b and 5.d, the posterior uncertainty ranges are generally comparable to or slightly narrower than the prior for medium-flow regimes. During low-flow conditions, however, the posterior bands are wider than the prior, indicating that incorporating SCA and GMB datasets introduces additional variability in streamflow predictions during low flow dominated periods, likely due to challenges in accurately constraining slow-response hydrological processes. For medium- and high-flow regimes, these datasets appear to modestly refine or maintain predictive uncertainty. In contrast, conditioning the model with isotopic data (Figure 5.f) results in a significant reduction in uncertainty, particularly during low-flow conditions. The posterior uncertainty for uncertainty during low-flow conditions.

**Figure 4.** Parameter distributions obtained by conditioning the model with streamflow observations recorded at the Nuxia station (prior PDF, black line) and by combining streamflow measures with: (i) snow cover area (posterior PDF, orange dashed line); (ii) glacier mass balance (posterior PDF, light blue dash-dotted line); and (iii) isotope concentrations (posterior PDF, green dotted line).

tainty range is substantially narrower than the prior, indicating improved model consistency in simulating low flow dominated periods.

To further enhance interpretability and provide a deterministic reference alongside the probabilistic representation, Figure 5 includes the mean simulated streamflow trajectories for both the prior and posterior distributions, in addition to the uncertainty bands and observed data. As evident from the figure insets and the FDCs, the prior and posterior means exhibit slight differences across all cases, with a more noticeable divergence of the posterior mean from the prior in the case of isotope conditioning.

These patterns of uncertainty reduction—particularly the distinct effect of isotopic data during low-flow periods—are also evident at the Yangcun and Nugesha stations (Figures S6 and S7 in the Supplementary Material), further supporting the conclusions presented above. To facilitate a direct comparison of streamflow magnitude across the three stations, Figure S8 provides time series and flow-duration curves (FDCs) of normalised streamflow for the period 2001–2010. It is important to note that any apparent loss of fit at the interior stations primarily reflects the fact that these sites (Yangcun and Nugesha) were not used for parameter selection. In other words, parameter sets were derived at Nuxia and then transferred without adjustment to these upstream gauges. As a result, the performance at Yangcun and Nugesha is affected by suboptimal parameter configurations. This loss of accuracy is consistent with previous findings (e.g., Khakbaz et al., 2012; Demirel et al.), which show that applying parameters calibrated at a single location to different location of the basin, without re-evaluating them locally, lead to reduced model performance.

# 3.4 Runoff component analysis







Figure 6 shows the CRC produced by different behavioral parameter sets. The boxplots illustrate the contributions of the four runoff components under the prior parameter set and the posterior parameter sets constrained by snow cover area, glacier mass balance, and isotope likelihoods. The contributions of subsurface runoff and rainfall surface runoff are similar, both accounting for approximately 40–45% of the total runoff (Figure 6.a and 6.b). In contrast, snowmelt surface runoff and glacier melt surface runoff contribute approximately 8% and 6%, respectively (Figure 6.c and 6.d). The estimated contributions of snowmelt and glacier melt are lower than some previous estimations in the study area (Boral and Sen, 2020; Lutz et al., 2014), but are close to some recent studies constraining the CRC estimation based on multiple datasets (Nan et al., 2025; Zhang et al., 2025; Chen et al., 2017).

The differences in the average CRCs among the parameter sets are relatively small, with variations generally below 3% for all four components. However, the inferences drawn from the different datasets reveal interesting patterns regarding uncertainty reduction. The prior leads to a wider distribution of contributions across all runoff components, reflecting higher uncertainty in the model predictions. Posterior parameter sets constrained by specific datasets help reduce this uncertainty to varying extents. Constraining the model with the likelihood of glacier mass balance leads to a significant reduction in the uncertainty of glacier melt surface runoff (Figure 6.d), as evidenced by the tighter interquartile range and fewer outliers in the box plot. This indicates that the GMB simulation provides strong constraints on glacier-related processes. In contrast, the snow cover area does not lead to a significant reduction in the uncertainty of snowmelt surface runoff (Figure 6.c). This is because SCA data only constraints the area of snow but does not provide much constraint on the volume of snow, as the snow area-volume

Figure 5. The 5–95% percentile prior, conditioned solely on streamflow, and posterior predictive uncertainty ranges for streamflow, calculated under different conditions: snow cover area (SCA), glacier mass balance (GMB), and isotopes (I). Panels (a), (c), and (e) show daily streamflow time series for the period 2010–2015, with insets highlighting low-flow dynamics; panels (b), (d), and (f) show flow duration curves for the full period 2001–2015. Streamflow data are presented in dimensionless form due to dissemination restrictions imposed by the data provider. A linear normalization,  $q^* = (Q - Q_{\min})/(Q_{\max} - Q_{\min})$ , is applied to the time series (panels a, c, e), while a logarithmic normalization,  $q^*_{\log} = (\log Q - \log Q_{\min})/(\log Q_{\max} - \log Q_{\min})$ , is used for the flow duration curves (panels b, d, f). In both cases,  $Q_{\min}$  and  $Q_{\max}$  refer to the minimum and maximum observed discharges at the Nuxia station.

relation is determined by a calibrated parameter. Notably, the isotope likelihood demonstrates a broader impact on reducing uncertainty across multiple runoff components. The boxplots for I show narrower distributions for subsurface runoff, rainfall

surface runoff, and snowmelt surface runoff, indicating that isotope simulation provides valuable constraints on both surface and subsurface hydrological processes.

The influence of each dataset on CRC uncertainties can be further illustrated by the result of sensitivity analysis, which evaluates the extent to which each performance metric is influenced by the contribution of each runoff component. To this end, Figure 7 presents the sensitivity of model performance metrics to the contributions of different runoff components, namely subsurface runoff  $(C_{ss})$ , rainfall surface runoff  $(C_{sr})$ , snowmelt surface runoff  $(C_{sm})$ , and glacier melt surface runoff  $(C_{sgm})$ . The sensitivity analysis evaluates the extent to which each performance metric—streamflow  $NSE_Q$ , snow cover area  $NSE_{SCA}$ , glacier mass balance  $VE_{GMB}$ , and isotope  $NSE_I$ —is influenced by the relative contribution of each runoff component to total streamflow.






The results indicate that streamflow performance  $\mathrm{NSE}_Q$  and snow cover area performance  $\mathrm{NSE}_{SCA}$  respond differently to variations in the contribution of individual runoff components. While  $\mathrm{NSE}_{SCA}$  remains largely insensitive to CRC variations, showing consistently high values across a wide range of runoff component contributions,  $\mathrm{NSE}_Q$  exhibits a more noticeable response. The scatterplots reveal that although streamflow performance remains relatively high ( $\mathrm{NSE}_Q > 0.8$ ) even when CRC deviates from its optimal value, there is a clear tendency for behavioral solutions to cluster towards an optimal CRC, indicating a degree of sensitivity. In contrast, glacier mass balance performance  $\mathrm{VE}_{GMB}$  shows strong sensitivity to glacier melt runoff  $C_{\mathrm{sgm}}$ , with  $\mathrm{VE}_{GMB}$  dropping significantly when  $C_{\mathrm{sgm}}$  exceeds approximately 10%. The most pronounced sensitivity is observed in the isotope performance metric  $\mathrm{NSE}_I$ , which responds to variations in multiple runoff components. The scatterplots reveal that  $\mathrm{NSE}_I$  declines markedly when the contributions of subsurface runoff  $C_{\mathrm{ss}}$ , rainfall runoff  $C_{\mathrm{sr}}$ , or snowmelt runoff  $C_{\mathrm{sm}}$  deviate from optimal values. In particular,  $\mathrm{NSE}_I$  decreases significantly from 0.4 to below 0.2 when the contributions of these components shift, indicating that isotopic simulations are much more sensitive to changes in runoff contributions compared to other performance metrics. This sensitivity underscores the importance of accurately quantifying the partitioning of different runoff components to achieve reliable isotope-based model predictions. Overall, the analysis highlights that  $\mathrm{VE}_{GMB}$  simulations are primarily sensitive to glacier melt runoff, whereas isotope-based simulations  $\mathrm{NSE}_I$  are more sensitive to a broader range of runoff components.

It is worth noting that, although NSE is not ideally suited to capture spatial features of snow cover dynamics, our analysis focuses on the catchment-integrated snow-covered area, for which NSE remains an informative metric to evaluate the agreement between observed and simulated temporal patterns of areal extent. In this regard, to better assess model performance, we provide the time series of observed versus simulated SCA in Figure S3, along with corresponding comparisons for glacier mass balance and isotopic signatures in Figures S4 and S5 of the Supplementary Material. These visualizations allow the reader to evaluate the temporal evolution and potential systematic biases for each variable, together with the associated posterior predictive uncertainty ranges for SCA, GMB, and isotopic data.

**Figure 6.** Boxplots showing the variability in the contributions of different surface runoff components under prior estimates conditioned solely on streamflow (Q) and posterior estimates conditioned on additional datasets: snow cover area (SCA), glacier mass balance (GMB), and isotopic data (I). Panel (a): Subsurface runoff; panel (b): rainfall surface runoff; panel (c): Snowmelt surface runoff; panel (d): glacier melt runoff.

Figure 7. Sensitivity of model performance metrics to runoff component contributions: streamflow  $NSE_Q$ , snow cover area  $NSE_{SCA}$ , glacier mass balance  $VE_{GMB}$ , and isotopes  $NSE_I$ , plotted against subsurface runoff ( $C_{ss}$ ), rainfall surface runoff ( $C_{sr}$ ), snowmelt surface runoff ( $C_{sm}$ ), and glacier melt surface runoff ( $C_{sgm}$ ). Each point represents a behavioral solution from the multi-objective calibration.

## 4 Discussion





Overall, the results presented in Section 3 highlight the differential value of auxiliary datasets in hydrological model calibration. While SCA and GMB provide insights into snow and glacier dynamics, they appear less effective in reducing streamflow uncertainty. Not only do the results prove that integrating multiple data sources within the Bayesian framework influences both streamflow simulation uncertainties and the computation of CRC components, but they also show varying effects depending on the type of dataset and runoff component considered, as discussed below.

## 4.1 Reducing Streamflow Model Uncertainty Using a Bayesian framework

The results of this study differ in another perspective from those of Di Marco et al. (2021), who observed a consistent relationship in snow-dominated basins between an increased likelihood of streamflow and snow cover area (SCA), alongside a reduction in streamflow uncertainty. In contrast, our findings do not show a comparable narrowing of streamflow uncertainty bands when applying the Bayesian filtering approach with snow and glacier parameters (Figure 5). This discrepancy suggests that the coupling between snow and glacier dynamics and streamflow performance is not straightforward, particularly in larger or more heterogeneous catchments.

As noted by Ruelland (2024), the potential for snow data to enhance streamflow simulation consistency and robustness depends on various factors, including hydro-climatic conditions, spatial variability, the modeling framework, and the accuracy of snow cover data (Hao et al., 2022) and input forcing (Raleigh et al., 2015). Factors such as catchment complexity, spatial heterogeneity, and structural uncertainties in the model, stemming from unresolved hydrological processes or oversimplified dynamics, likely contribute to the persistence of wide uncertainty ranges. In contrast, isotopic likelihoods effectively constrain the parameter space, resulting in improved simulation performance and reduced uncertainty bands, particularly during low-flow conditions. This finding confirms the ability of isotopic data to capture key hydrological processes, such as groundwater-surface water mixing and subsurface flow dynamics, which are especially influential during low-flow periods (Jasechko and Taylor, 2015), where seasonality plays a critical role (Bierkens et al., 2001; Birkel et al., 2009).

The influence of hydrological processes seasonality on the effectiveness of likelihoods is demonstrated by the sharpness polar plot (Figure 8). This figure illustrates the sharpness ranges for posterior likelihoods conditioned on SCA, GMB, and I datasets throughout the year. A maximum SH value of 0.34 was observed for isotopes on March 16, 2008, while the maximum SH values for SCA and GMB were 0.19 on April 30, 2009, and 0.16 on June 10, 2009, respectively. These results highlight the effectiveness of isotopic likelihoods during winter and early spring, with sharpness values remaining consistently narrow and never dropping below zero, a period when the model indicates a predominance of contributions from the subsurface flow component. In contrast, SCA and GMB likelihoods achieve their sharpness peaks during spring and early summer, coinciding with periods of rapid snowmelt and glacier runoff. This pattern underscores the importance of integrating SCA and GMB likelihoods for capturing high-flow dynamics and highlights the need to further develop these datasets to enhance their effectiveness in constraining streamflow uncertainty during these critical periods.

At this point, it is important to recall that sharpness refers to the degree of concentration of the ensemble simulations where a sharper ensemble has a narrower spread, indicating higher predictive confidence (Gneiting et al., 2007). However, increased sharpness does not necessarily translate into improved reliability. In our case, the inclusion of isotopic data led to a more constrained ensemble, resulting in a sharper posterior distribution of streamflow simulations. While this outcome reflects the stronger constraining power of isotopic information, it also increased the likelihood that observed values fall outside the model's predictive bounds, thereby reducing the containing ratio (CR). Compared to calibrations using snow-covered area (SCA) or glacier mass balance (GMB), the sharper ensemble derived from isotope-informed calibration was less able to fully capture observed variability. This illustrates the classic trade-off between predictive confidence and reliability — in other words, between sharpness and containing ration in probabilistic modeling (Beven and Binley, 1992), and emphasizes the need to balance these aspects when evaluating ensemble-based hydrological simulations.

These results confirm that isotopic data are highly effective in reducing model uncertainty by providing independent constraints on flow partitioning and subsurface processes. However, to translate this enhanced internal consistency into improved predictive coverage, future research should explore model structural refinements that better align sharpness with CR. Furthermore, these findings illustrate both the potential and the limitations of Bayesian inference in simultaneously capturing fast surface runoff and slower subsurface dynamics. Although sharpness values demonstrate its capacity to constrain parameter uncertainty across diverse hydrological processes, alternative calibration strategies, such as multi-objective weighted optimization, may offer additional improvements in streamflow simulation accuracy (He et al., 2019). Still, the sensitivity of model outputs to weight selection necessitates careful application (Tong et al., 2021, 2022). Finally, the interplay between likelihood functions underscores the metric-dependent nature of parameter uncertainty reduction and the value of integrating multiple complementary evaluation criteria during calibration (e.g., Fenicia et al., 2018; Majone et al., 2022).

These results also point to the need for improved coupling and integration of individual model components. Such integration would allow for better exploitation of the strengths of each dataset and enhance the Bayesian framework's capability to constrain parameter ranges across diverse hydrological conditions. By addressing these structural connections and leveraging synergies between complementary metrics, the Bayesian framework's potential to optimize parameter calibration and improve predictive accuracy can be fully realized.

In this context, the posterior mean streamflow, especially in the isotope-conditioned simulations, fails to consistently outperform the prior mean streamflow in reproducing the observed discharge, despite exhibiting narrower uncertainty bands in some streamflow regimes (see Section 3). This deterioration in deterministic skill is not unexpected. Previous studies (e.g., Vrugt and Sadegh, 2013; Botto et al., 2018) have shown that reducing ensemble spread does not automatically lead to improved agreement with observations. Structural model deficiencies and varying accuracy of input data sources (i.e., SCA, GMB, and I) may introduce systematic posterior bias, since the conditioning step attempts to compensate for processes that are poorly captured by the model or affected by different levels of uncertainty (Beven and Freer, 2001; Chowdhury and Sharma, 2007).

It is important to emphasize that the ensemble mean does not correspond to the best-performing simulation in terms of NSE, and may smooth out dynamic features that are better reproduced by individual ensemble members. Moreover, the goal of the data-conditioning approach is not to maximize deterministic skill, but rather to reduce predictive uncertainty by constraining

the prior ensemble: the shift from prior to posterior aims at narrowing the uncertainty bands of the streamflow simulations, even at the cost of some loss in individual accuracy (Beven, 2006).

## 4.2 Runoff Component Uncertainty








The GMB dataset effectively reduces uncertainty in glacier melt surface runoff simulations (Figure 6.d), emphasizing its value for improving model constraints in glacier-dominated systems. This finding aligns with previous studies highlighting the importance of incorporating GMB data to enhance streamflow predictions in such catchments (Stahl et al., 2008; O'Neel et al., 2014; Yang et al., 2024). However, this reduction in uncertainty does not always translate into improved streamflow predictions at the basin scale. The effectiveness of the Bayesian framework in reducing uncertainties depends on the proportion of runoff attributed to glacier melt processes. Consequently, even when glacier-related dynamics are well constrained by GMB data, their contribution to reducing overall streamflow prediction uncertainty may be limited in basins where other processes dominate. This underscores the importance of considering basin scale and dominant runoff processes when selecting datasets for hydrological modeling.

Similarly, SCA datasets provide valuable constraints on snowmelt surface runoff (Figure 6.c) but have a more limited impact on reducing streamflow uncertainty. This may be due to the spatial and temporal resolution limitations of SCA datasets (Di Marco et al., 2020), or because the snowmelt contribution to total runoff is relatively minor in large basins compared to other components, such as subsurface runoff and rainfall surface runoff. Furthermore, uncertainties in the timing and rate of snowmelt, which are critical for runoff generation, may not be fully captured by remotely sensed SCA data (Andreas Juergen Dietz and Dech, 2012). This limitation is particularly relevant in basins with complex snow dynamics, where snow cover depletion varies significantly across different elevation bands and time periods (Molotch and Margulis, 2008).

In contrast, isotopic data stand out for their ability to reduce uncertainty across multiple runoff components, particularly during low-flow conditions. By tracing water sources and pathways, isotopic tracers provide critical insights into subsurface and groundwater contributions, which are difficult to capture with traditional datasets (Birkel et al., 2015). Isotopic tracers, such as oxygen-18 ( $\delta^{18}$ O) and deuterium (D), are widely used to distinguish between recent precipitation, snowmelt, and groundwater contributions to streamflow, improving the calibration of hydrological models (Jasechko, 2019). Our results show that the isotope data does not reduce the uncertainty of glacier melt runoff, because the model assumes that glacier melt generate runoff directly through surface pathway, not involved in the surface-subsurface runoff partitioning, the aspect for which isotopic data are most useful. The results suggest that incorporating isotopic data into hydrological models can help reduce uncertainties related to water source contributions and flow pathways, particularly in catchments with complex surface-subsurface interactions. Such benefit comes from the significant divergence in the isotope signatures of surface and subsurface runoff, i.e., a much lower temporal variability of groundwater isotope compared to surface runoff because of the long travel time.

These differences in the influence of datasets underscore the importance of selecting appropriate data sources based on the specific hydrological processes and uncertainties that need to be addressed in a given catchment. For example, GMB data should be prioritized in glacier-fed basins to improve predictions of glacier melt runoff (Huss and Hock, 2015), whereas

isotope data can provide valuable constraints on multiple runoff components, particularly in catchments with diverse flow generation processes (Rodgers et al., 2005; Birkel et al., 2011). The integration of multi-source datasets can help reduce model uncertainties more effectively than relying on a single dataset (Beven, 2006), resulting in more robust predictions of water availability and streamflow variability under changing climatic conditions (Borriero et al., 2023).

## 4.3 Limitations






This study systematically evaluates the value of snow cover area, glacier mass balance, and isotopes in reducing model uncertainties. Results highlight the critical role of isotope data in improving low-flow simulations and runoff component separation. However, several limitations persist. First, while streamflow simulations achieve NSE values up to 0.9, peak flows are consistently underestimated, likely due to inaccuracies in precipitation forcing data (Jiang et al., 2022; Xu et al., 2017). Metrics for SCA and isotope simulations remain around 0.5, indicating potential for further optimization. Second, the model structure is rather simplified when conceptualizing processes such as groundwater and snow/ice melting. In specify, only two subsurface layers (u-zone and s-zone in the model) are defined, and the subsurface outflows are simulated as a sum. Only the shallow groundwater processes are considered, only occurring within each REW, which is unable to adequately describe subsurface processes in the TP, where deep interbasin groundwater pathways exist (Chen et al., 2025). Meanwhile, the simple degreeday factor method was used to simulate the melting processes, to make the model adequately efficient for subsequent GLUE analysis. These modules can be improved to strengthen the physical basis of the model. Third, as this analysis is based on a single case study in a specific region, its broader applicability is uncertain. Unlike prior studies (Di Marco et al., 2021; Tong et al., 2021), snow and glacier datasets did not significantly enhance model performance here, suggesting the need to clarify the conditions under which such data prove most beneficial.

Despite these challenges, the study underscores the importance of employing multiple datasets to constrain hydrological models. Although snow and glacier datasets alone may not substantially improve streamflow simulations, they are essential for ensuring model reliability in capturing key processes. Isotope data, in particular, effectively constrain surface and subsurface runoff separation due to the low variability in groundwater isotopic composition (Nan et al., 2024; McGuire and McDonnell, 2006), reducing low flow uncertainties and enhancing model robustness.

#### 5 Conclusions

This study provides new insights into reducing uncertainty and equifinality in the hydrological modeling of large mountainous catchments by integrating multiple auxiliary datasets within a Bayesian framework. By systematically comparing the contributions of snow cover area (SCA), glacier mass balance (GMB), and isotopic tracers, we demonstrate how these datasets distinctly improve model performance across various flow regimes.

A critical conclusion drawn from this research is the unique advantage of isotopic data in reducing model uncertainty during low-flow periods. The isotopic likelihood has shown to be more effective in constraining subsurface flow contributions and groundwater-surface water interactions, resulting in narrower uncertainty ranges for streamflow predictions under low-

flow conditions. This finding underscores the critical role of isotopic tracers in improving the representation of slow-response hydrological processes, which are essential for the mitigation of drought and sustainable management of water resources in mountainous regions. In contrast, the SCA and GMB datasets were found to be more effective in capturing rapid surface dynamics, such as snowmelt and glacier melt processes. However, their contributions to reducing streamflow uncertainty were limited, particularly during low-flow conditions. This discrepancy highlights the need for multi-objective calibration approaches that balance the trade-offs between rapid surface responses and slow subsurface processes.

Our results also reveal the differential impact of each dataset on the contributions of runoff components. The glacier mass balance likelihood significantly reduces uncertainty in glacier melt surface runoff, whereas isotopic data provide broader constraints across multiple runoff components, including subsurface runoff, rainfall surface runoff, and snowmelt surface runoff. These differences emphasize the importance of selecting appropriate datasets based on the dominant hydrological processes in a given catchment.




The study further highlights the limitations of current Bayesian frameworks in fully leveraging the complementary strengths of auxiliary datasets. While Bayesian approaches are effective in reducing parameter uncertainty and improving model calibration, the persistent wide uncertainty ranges for streamflow predictions indicate the need for improved coupling and integration of individual model components. Enhancing these structural connections within the modeling framework could allow for better exploitation of multi-source datasets, ultimately improving predictive accuracy across diverse hydrological conditions.

In conclusion, our findings stress the importance of incorporating multi-source datasets in hydrological modeling to achieve robust performance across different flow regimes. The integration of isotopic tracers, snow cover, and glacier mass balance data within a Bayesian framework offers a promising pathway to reduce uncertainty and enhance the understanding of streamflow variability in large mountainous catchments. Future research should focus on developing more advanced coupling methods that account for the complex interplay between cryospheric and subsurface processes, as well as exploring the potential of multi-objective weighted calibration approaches to further improve model reliability.

Figure 8. Polar plots showing the daily sharpness band computed from the maximum and minimum sharpness values across the years 2010–2015. The shaded regions represent the range of sharpness variability for each day of the year, while the solid black line indicates the reference level at zero sharpness. The subplots illustrate the sharpness calculated under different conditioning:  $cLH(p_i|Q,SCA)$  (a),  $cLH(p_i|Q,GMB)$  (b), and  $cLH(p_i|Q,I)$  (c).

Code availability. The code of THREW-T model used in this study are available from the corresponding author (ny1209@qq.com)

Data availability. Data sets are publicly available as follows: DEM (http://www.gscloud.cn/sources/details/310?pid=302, last access: 1 January 2019, Geospatial Data Cloud Site, 2019), CMFD (China Meteorological Forcing Data,

https://doi.org/10.11888/AtmosphericPhysics.tpe.249369.file, Yang and He, 2019), glacier inventory data (https://doi.org/10.3972/glacier.001.2013.db, Liu, 2012), glacier elevation change data (https://doi.org/10.6096/13, Huggonet et al., 2021), NDVI (https://doi.org/10.5067/MODIS/MODIS/MOD13A3.006, Didan, 2015), LAI (https://doi.org/10.5067/MODIS/MOD15A2H.006, Myneni et al., 2015), HWSD (Harmonized World Soil Database,

https://data.tpdc.ac.cn/zh-hans/data/3519536a-d1e7-4ba1-8481-6a0b56637baf/?q=HWSD, last access: 1 January 2019, He, 2019). The simulated streamflow, snow cover area, glacier mass balance and isotope data produced by the behavioral parameters and the 5th-95th streamflow uncertainty bands for both the prior and posterior ensembles are provided in Zenovo at https://zenodo.org/records/15605202; https://zenodo.org/records/16634369). The datasets not publicly available are referred to in the main text (Chen et al., 2018; Liu et al., 2007).



Competing interests. At least one of the (co-)authors is a member of the editorial board of Hydrology and Earth System Sciences.

Acknowledgements. During the preparation of this work the author(s) used ChatGPT in order to improve language and readability of the original draft, with caution. After using these tools, the author(s) reviewed and edited the content as needed and take(s) full responsibility for the content of the publication This study has been supported by the National Natural Science Foundation of China (grant no. 52309023) and the Fund Program of State Key Laboratory of Hydroscience and Engineering (sklhseTD-2024-C01). Diego Avesani acknowledges the Italian Ministry of Education, Universities and Research (MUR), in the framework of the project DICAM-EXC [Departments of Excellence 2023–2027, grant L232/2016]; and the support from the European Union – FSE-REACT-EU, PON Research and Innovation 2014–2020 DM1062/2021; PAT Project Drought Indicators - CUP C97G22000460003. The open-access publication of this study is supported by the Fund Program of State Key Laboratory of Hydroscience and Engineering (sklhseTD-2024-C01).

Author contributions. DA was responsible for conceiving the study; developing the methodology; acquiring funding; carrying out the investigation; developing the software; preparing, reviewing, and editing the manuscript; and supervising the study. YN conceived the study, contributed to developing the software, carrying out the investigation, creating the figures, curating the data, and reviewing and editing the manuscript; FT reviewed and edited the manuscript, and supervised the study.

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
