# Peer review of "Reducing Hydrological Uncertainty in Large Mountainous Basins: The Role of Isotope, Snow Cover, and Glacier Dynamics in Capturing Streamflow Seasonality"

_EGUsphere, 2025_

## Referee Comment (RC2)

**Review for "Reducing Hydrological Uncertainty in Large Mountainous Basins:**

**The Role of Isotope, Snow Cover, and Glacier Dynamics in**

**Capturing Streamflow Seasonality"**

**General comments:**

This manuscript focuses on evaluating the value of snow cover area, glacier mass balance, and isotopes in reducing uncertainty and equifinality of hydrological modeling in a large mountainous basin in the Tibetan Plateau. The Bayesian approach and GLUE method are adopted to investigate the research questions. The research topic aligns with the journal scope and the research findings are potentially useful for the readers. I have a few concerns regarding the modeling procedure, the details of the input data, and the interpretation of the results before the paper being accepted for publication.

Additionally, one thing I noticed here is that the time-series simulated and observed discharge does not have a y-axis (Fig.5), which is present on purpose due to data dissemination restrictions mentioned in the caption. However, this is not possible for readers to understand the model performance, and the magnitude of the simulated and observed discharge. A manuscript avoiding showing y-axis of time-series discharge plot in the results could potentially conflict with the basic principle of open science of HESS/Copernicus journals.

**Specific comments:**

**Modeling perspective:**

1.  The subsurface is overly-simplified represented in the model. The subsurface flow generates from the model is composed of the subsurface lateral flow ("interflow") in the unsaturated zone and the baseflow from groundwater to surface water in the saturated zone. These two subsurface flow components are simulated as a sum (L105 and Fig.1). It is thus not possible to conclude the role of groundwater in contributing to the streamflow and the groundwater-surface water interactions. The subsurface lateral flow can be high and not negligible in such large mountainous basin (>$2*10^5$ km$^2$). It is recommended to be cautious in interpreting and concluding the result regarding the baseflow. All mentioning of groundwater baseflow in the manuscript actually refer to the subsurface flow, i.e. the sum of both unsaturated and saturated zone, e.g. on L134, it is subsurface flow, but not baseflow. The presented modeling approach is not able to investigate groundwater alone.
2.  Regarding the modeling, are the spatial zones delineated the same for both the surface and subsurface? (this could potentially fragment the aquifers located at the boundaries). Is the subsurface flow allowed to cross the delineated boundaries? The conceptualization of the subsurface processes in the model potentially limits the ability of the model for investigating the surface-subsurface interactions. The model limitation should be clearly discussed in Section 4.3.
3.  There are 4 discharge stations, but only the results at Nuxia Station are presented. The results for the other stations should be presented in the Supplementary Information. The authors

should also clarify if the conclusions achieved at Nuxia Station are held the same as the other three stations.

4. Does glacier melt contribute to groundwater recharge? Or is it assumed that all glacier melt goes into streamflow? This assumption should be clear in the text as well.

5. The simple degree-day-factor methods are used to solve snowmelt and glacier melt. Glacier mass balance is estimated with a simple volume-area scaling factor approach. The limitations of these adopted simple approaches for solving snow and glacier melts should be discussed in terms of modeling limitations.

6. L213: how are the Pareto fronts defined? Please justify this threshold used to show the Pareto fronts in Figure 3 and the conclusions obtained from this result relating to this threshold.

7. L248-249: Why does the KKA shows a noticeable convergence, but not KKD? They both are parameters that control the subsurface runoff outflow rates. Please clarify this point.

**Data perspective:**

Limited information is provided on the input data of this study. This could hamper the readers to interpret the results.

1. L79: The four river gauging are only given by names and no other information and data are available. It is recommended to provide details on the coordinates and elevations of the four river gauging stations in this mountainous basin, also their observed periods, frequency, and measurement method. Any observations errors/failures in the winter low flow and high flow periods? These details are important to interpret the observed and simulated discharge.

2. Section 2.1: What is the modelling period? Please detail the start and end dates of the meteorological data sets and the modelling period. Also add details on which years of DEM, land use data, soil data, snow cover, and glacier data are used in this modelling study.

3. L79-93: Are the gridded meteorological satellite data corrected with in-situ station data? How are the different resolutions of various types of gridded spatial data used in the hydrological model? Please provide details on this.

4. The description of the streamflow sampling is very vague, which is simply stated as "Grab samples of stream water were collected in 2005 at four stations..". Please provide details on how many samples and in which months the samples were collected. Do the authors have the precipitation (rainfall, snow) isotopes in the same year (2005) or in a different year (2008)? Using streamflow and precipitation isotope data of different years in the same model can be inappropriate.

5. How is the precipitation tracer estimated for rainfall and snow individually? This needs to be clarified in the manuscript.

**Interpretation of the results:**

1. L256-259: The SCA shows a higher influence on the posterior distribution of T0 than the GMB, which does not show the strongest influence as the authors interpreted. Could the authors please clarify why they see GMB as the strongest from this result figure (Fig.4j)?

2. L306-308: The isotope data have increased uncertainty of the simulated glacier melt runoff (Fig.6d), but they are helpful to constrain other surface runoff components (rainfall runoff, snowmelt). Please clarify this result.

3. L272-277: Including the isotope data leads to a decreased containing ratio. This means a significant under capture of the extremely low and high streamflow. Why including isotope data has decreased the streamflow simulation performance? Please clarify this result.

**Technical corrections:**

L8: It would be helpful to mention which type of hydrological model the THREW-T is in the abstract. i.e. fully-distributed, semi-distributed, or conceptual?

L78: km$^2$ should be straight upright, not italic. Please correct all formats of the units for similar cases.

L80, L82: Please add years between which the mean annual precipitation and mean annual temperature are calculated.

L100: distributed -> semi-distributed?

L104: what is bare zone? Bare soil, bare rock?

Figures 2 and 4: avoid using red and green colors together in the same figure to allow readers with colour vision deficiencies to correctly interpret your findings.

Table 1: table caption should be on top of the table.

L165: Please correct the formats of Equations 1-6 by following the journal guideline. e.g. the NSE should be straight upright, not italic. The text subscription should be straight upright as well.

L210-211: NSE, VE should be straight upright, not italic. Please check the format of all such mentioning.

L266-267, 281: Please remove the parentheses around the Section and Figure numbers, and correct all such mentioning in the manuscript.

Figure 4 caption i) covered area -> snow covered area.

---

## Author Comment (AC1)

**Reviewer 1**

The manuscript presents a hydrological modeling study in a glacier-influenced catchment. The work explores the value of auxiliary datasets, namely water isotope composition, snow cover area, and glacier mass balance in model calibration in a GLUE framework. The model structure allows tracer simulations and comparison with spatially variable datasets. The works finds different datasets have more power in model calibration in different hydrological seasons: isotopes during baseflow, and snow and glacier related observations during the melt period.

I liked the systematic approach for including model validation datasets of very different origin to model evaluation scheme. The GLUE uncertainty analysis framework for the work is in my judgement valid. The overall approach the authors develop to explore parameter sensitivity to model validation objectives and stream source water contribution are in my opinion of interest to the community. I recommend the work to be published after addressing my comments below:

MAJOR COMMENTS

I'd like to see better presentation of the stable water isotope data. You have only isotope data of the streamflow validation, it remains unclear how representative the input precipitation data is of the catchment. Do any of the references cited for the model development have any comparison data for simulated precipitation, snow or groundwater isotope composition? Having even cursory validation of the simulated isotope composition in different model compartments (snow, glacial melt, groundwater mainly) in the would give more credibility that the streamflow isotopes are correctly simulated and informative for the right reasons. On that note, I'd like to see a figure of the stream isotope data and model simulation fit to stream isotopes.

**Response:**
We will add more descriptions about the isotope of various water bodies.
● Precipitation: Our previous evaluation of isoGSM (Nan et al., 2021) indicated that it can effectively capture the seasonal variation in precipitation $\delta^{18}O$, but exhibited a systematic overestimation bias in the study region and performed relatively poorly in accurately capturing the isotope signature of specific events (see the Figures 1 and 2). We adopted the corrected isoGSM product from Nan et al. (2022) as the input data, in which the bias of isoGSM was adjusted based on a linear regression with altitude. Note that the corrected isoGSM directly incorporates measured precipitation $\delta^{18}O$ data for locations and dates with observations, so comparing the corrected isoGSM with measured data is not meaningful. Instead, we only show the relationship between the original isoGSM and measurements to illustrate the capability of the original isoGSM in simulating precipitation $\delta^{18}O$.

- Glacier melt: The $\delta^{18}O$ of glacier meltwater was calculated using the offset-parameter method, in which the glacier-melt $\delta^{18}O$ was assumed to be temporally constant and 5‰ lower than the weighted average of local precipitation $\delta^{18}O$. The value of the offset parameter (5‰) was estimated from the data collected by Boral and Sen (2020).
- Groundwater: We were not able to collect groundwater samples for isotope validation. Nonetheless, the characteristics of groundwater $\delta^{18}O$ are clear: it exhibits much lower temporal variation than precipitation or streamflow, due to the long water travel time.
- Snowmelt: We were also not able to collect snowmelt water samples for isotope validation. So the parameters constrained by isotope didn't lead to significant difference in snow simulation and the estimation of snowmelt runoff. The snow simulation is mainly constrained by the snow cover area data.
- Streamflow: Yes. We will add a figure to show the simulation and observation of stream water $\delta^{18}O$.

[Figure]

*Fig. 1: Temporal variations in the precipitation $\delta^{18}O$ derived from observation and isoGSM data at four stations.*

[Figure]

*Fig. 2: The scatter diagrams between the isoGSM and measured isotope data at four stations.*

The fractions for snowmelt surface runoff and glacier surface runoff seem low to me. Can you provide comparison with fractions found in other montanous snow and glacier influenced sites? Quite often end-member mixing analysis fraction estimations are done for three end members: snow, rain and glacial melt. In your model analysis groundwater is explicitly considered as a component, but isotopically it is essentially composed of rain, snow and glacial melt. This in my opinion creates a bit of confusion, and makes the glacial and snow melt seem less important for the regions water resources. I don't think there is an error in your analysis, but would be good to clarify the concepts further, to make your results more relatable to other literature.

Response:
- There are two common ways to define runoff components. One is based on water source, describing where the water originates; under this definition, the three end-members are rainfall, snowmelt, and glacier melt. The other is based on the runoff-generation pathway, describing how water produces runoff; here, the two end-members are surface runoff and subsurface runoff. In both cases, the sum of component contributions equals one. Because groundwater is also important, including all components would require reporting two separate sets of results (e.g., rainfall 80%, snowmelt 10%,

glacier melt 10%; surface runoff 40%, subsurface runoff 60%), which can be confusing. Therefore, in this study we combined these definitions and defined four runoff components.

- The contribution of runoff component is highly dependent on the calculation definition and the dataset used for model validation. In our result, the low contribution of snow and glacier runoff is partly due to our component definitions and the correspondingly high share of subsurface runoff. The contributions of snowmelt and glacier meltwater in the study area are highly uncertain, ranging from less than $5\%$ to over $30\%$. Nonetheless, studies using snow and glacier data to validate the model—thereby enhancing reliability—have all estimated relatively low contributions. For example, Chen et al. (2017) estimated snowmelt and glacier-melt contributions as $10.6\%$ and $9.9\%$ (ratios defined as SM/Q and GM/Q), whereas Zhang et al. (2025) reported corresponding values of $6.0\%$ and $6.2\%$ (ratios defined as $SM/(RF+SM+GM)$ and $GM/(RF+SM+GM)$). When calculated in the same manner, our results closely match these estimates.

MINOR COMMENTS

L12: I perceive GW-SW interactions as specific water exchange processes between surface and subsurface water. As you don't really delve deeper into GW-SW interactions in your simulations, I'd propose that you stick with talking only about baseflow, not GW-SW interactions (which baseflow generation if of course a manifestation of)

Thank you for this clarification. We agree with your observation and will avoid using the term "groundwater–surface water interactions" in the revised manuscript, as our analysis does not explicitly address these processes. Instead, we will refer to subsurface flow, which is more consistent with the scope of our simulations and aligns with Reviewer 2's comments

L54-L66: seems like the research questions are to some extent repeated. Suggest to review and rewrite more concisely.

Thank you for the suggestion. We acknowledge the redundancy and will revise the paragraph to make the research questions more concise and focused in the revised manuscript.

L110: Do you think snow sublimation would be a significant flux in your region, possibly influencing the snow storage and isotope composition of the snowpack consequently snow melt?

Some studies indicated that due to the wet condition in the YTR basin, the sublimation losses are relatively small, accounting for only 2-3% of the annual snowfall (Lutz et al., 2016, Khanal rt al., 2021). Consequently, our model doesn't consider the influence of snow sublimation, as some other modeling studies in this basin (e.g., Chen et al., 2017, Sun at al., 2024).

L213: not clear how the simulations comprising the pareto front (red markers in

are selected. seems like the number of the included simulations is fairly low, around 15.

Thank you for your observation. The red markers correspond to simulations on the Pareto front, identified in the bi-objective space shown in each panel. The relatively small number of red points (~15) reflects the fact that only a limited subset of simulations are non-dominated with respect to both objectives. We emphasize that the Pareto front is computed over the entire simulation ensemble, independently of any behavioral classification. The red points should therefore not be interpreted as behavioral simulations, but rather as Pareto-optimal solutions based solely on the two performance metrics shown. This behavior is consistent with what was reported by Di Marco et al. (2021), who found that the number of Pareto-optimal simulations was substantially lower than the number of behavioral ones, highlighting how multi-objective trade-offs can lead to highly selective optimal subsets. We will revise the manuscript and the figure caption to make this distinction clearer.

L238: can you further explain where the prior parameter distributions in Fig.4 comes from. Is it the parameters with >0 NSE for streamflow?

We agree that the origin of the prior parameter distributions shown in Fig. 4 requires further clarification. The prior parameter distributions are derived from model parameter sets that resulted in a positive Nash-Sutcliffe Efficiency (NSE > 0) for streamflow. This filtering step ensures that only behaviorally plausible parameterizations—those capable of reproducing streamflow dynamics to a reasonable degree—are included in the prior. We will revise the manuscript to explicitly state this criterion in the main text, and we will update the caption of Fig. 4 accordingly for clarity.

L304: I don't fully understand why the sensitive LL parameter does not manifest in the snowmelt fraction.

From a water balance perspective, the contribution of snowmelt is primarily governed by the fraction of snowfall in total precipitation, which depends on the temperature threshold used for rainfall/snowfall partitioning. While the LL parameter mainly affects the spatial extent of snow cover — and thus the spatial and temporal distribution of snowmelt — its influence on the total amount of snowmelt remains limited. We will clarify this aspect more explicitly in the revised manuscript.

L307: the narrower ranges for isotope simulations are not evident visually compared to the Q simulations. Would any statistical test either looking for differences in central values or variability in the distributions be helpful in identifying the differences?

We will provide some quantitative results to illustrate this.

L310: incomplete sentence?

Thank you for pointing this out. We will correct the manuscript accordingly.

L341-343: not very clear how successful the snow cover extent simulations are in the first place. The NSE metric is not very intuitive for snow cover extent variable. If for example the extent in area does not quantify, if the snow cover is simulated in the correct location. Similarly as requested for the isotopes, can you provide the timeseries of observed vs simulated snow cover extent to identify and discuss some potential some biases.

Thank you for this comment. We agree that NSE, while commonly used, may not fully capture spatial aspects of snow cover dynamics. Our analysis focuses on the catchment-integrated snow-covered area (SCA), where NSE remains an informative metric for evaluating the agreement between observed and simulated temporal patterns of areal extent. To better illustrate model performance, we have included the time series of observed vs. simulated SCA in Figure 3, along with corresponding comparisons for glacier mass balance (GMB) and isotopic signatures in Figures 4 and 5. These figures allow the reader to visually assess the temporal dynamics and potential biases for each variable. As detailed in Sections 3.3 and 2.2, the figures also represent the posterior predictive uncertainty ranges for streamflow, SCA, GMB, and isotopic data. These will be provided as part of the Supplementary Material of the revised manuscript. We believe these additional results will help clarify how successful the simulations are in reproducing the observed seasonal and interannual variability.

[Figure]

*Fig. 3. Observed Snow Cover Area and posterior model ensemble LH(p_i | Q, SCA), based on joint conditioning with streamflow and snow cover area information.*

[Figure]

*Fig. 4. Comparison between observed δ¹⁸O and the model posterior ensemble LH(pᵢ | Q, I), conditioned on streamflow and isotope information.*

[Figure]

*Fig. 5. Observed glacier mass balance and posterior model ensemble LH(pᵢ | Q, GMB), based on joint conditioning with streamflow and mass balance information.*

**Bibliography:**

Boral, S., & Sen, I. S. (2020). Tracing 'Third Pole' ice meltwater contribution to the Himalayan rivers using oxygen and hydrogen isotopes. Geochemical Perspectives Letters, 13, 48–53. https://doi.org/10.7185/geochemlet.2013

Chen, X., D. Long, Y. Hong, C. Zeng, and D. Yan (2017), Improved modeling of snow and glacier melting by a progressive two-stage calibration strategy with GRACE and multisource data: How snow and glacier meltwater contributes to the runoff of the Upper Brahmaputra River basin?, Water Resour. Res., 53, 2431–2466, doi:10.1002/2016WR019656

Di Marco, N., Avesani, D., Righetti, M., Zaramella, M., Majone, B., & Borga, M. (2021). Reducing hydrological modelling uncertainty by using MODIS snow cover data and a topography-based distribution function snowmelt model. Journal of Hydrology, 599, 126020. https://doi.org/10.1016/j.jhydrol.2021.126020

Lutz, A. F., Immerzeel, W. W., Kraaijenbrink, P. D. A., Shrestha, A. B., & Bierkens, M. F. P. (2016). Climate change impacts on the Upper Indus hydrology: Sources, shifts and extremes. PLOS ONE, 11(11), e0165630. https://doi.org/10.1371/journal.pone.0165630

Khanal, S., Lutz, A. F., Kraaijenbrink, P. D. A., van den Hurk, B., Yao, T., & Immerzeel, W. W. (2021). Variable 21st century climate change response for rivers in High Mountain Asia at seasonal to decadal time scales. Water Resources Research, 57, e2020WR029266. https://doi.org/10.1029/2020WR029266

Nan, Y., He, Z., Tian, F., Wei, Z., & Tian, L. (2021), Can we use precipitation isotope outputs of isotopic general circulation models to improve hydrological modeling in large mountainous catchments on the Tibetan Plateau? Hydrology and Earth System Sciences, 25(12), 6151–6172. https://doi.org/10.5194/hess-25-6151-2021

Nan, Y., He, Z., Tian, F., Wei, Z., & Tian, L. (2022), Assessing the influence of water sampling strategy on the performance of tracer-aided hydrological modeling in a mountainous basin on the Tibetan Plateau. Hydrology and Earth System Sciences, 26(15), 4147–4167. https://doi.org/10.5194/hess-26-4147-2022

Sun, H., Yao, T., Su, F., Yang, W., and Chen, D.: Spatiotemporal responses of runoff to climate change in the southern Tibetan Plateau, Hydrol. Earth Syst. Sci., 28, 4361–4381, https://doi.org/10.5194/hess-28-4361-2024, 2024.

---

## Author Comment (AC2)

**Reviewer 2**

**General comments:**
This manuscript focuses on evaluating the value of snow cover area, glacier mass balance, and isotopes in reducing uncertainty and equifinality of hydrological modeling in a large mountainous basin in the Tibetan Plateau. The Bayesian approach and GLUE method are adopted to investigate the research questions. The research topic aligns with the journal scope and the research findings are potentially useful for the readers. I have a few concerns regarding the modeling procedure, the details of the input data, and the interpretation of the results before the paper being accepted for publication.
Additionally, one thing I noticed here is that the time-series simulated and observed discharge does not have a y-axis (Fig.5), which is present on purpose due to data dissemination restrictions mentioned in the caption. However, this is not possible for readers to understand the model performance, and the magnitude of the simulated and observed discharge. A manuscript avoiding showing y-axis of time-series discharge plot in the results could potentially conflict with the basic principle of open science of HESS/Copernicus journals.

We thank the reviewer for raising this important point. We fully acknowledge that omitting the y-axis from the time-series discharge plots limits the reader's ability to assess the absolute magnitude of both simulated and observed flows. We also recognize that this may appear to be at odds with the open science principles promoted by HESS and Copernicus journals. However, the decision to exclude the y-axis was made to comply with data dissemination restrictions imposed by the data provider, who explicitly prohibits the publication of absolute discharge values. We are aware of the implications of this limitation and will revise the manuscript to address it as transparently as possible. Specifically, we will: (i) add a normalized y-axis to enable relative performance assessment; (ii) include a note explaining how access to the original data can be requested under the provider's terms; and (iii) explicitly discuss this limitation in the manuscript, including a clear justification and a reflection on its implications for reproducibility. We note that similar cases—where normalization or partial data concealment was necessary due to access restrictions—have been accepted in previous HESS publications, provided they were accompanied by proper documentation and acknowledgment of the limitations (e.g., Nan et al., 2021 and Nan and Tian 2024 ). We are, of course, open to following the editor's guidance on how to best address this issue while respecting both the journal's policies and the constraints imposed by the data provider.

**Specific comments:**
**Modeling perspective:**
The subsurface is overly-simplified represented in the model. The subsurface flow generates from the model is composed of the subsurface lateral flow ("interflow") in the unsaturated zone and the baseflow from groundwater to surface water in the saturated zone. These two subsurface flow components are simulated as a sum (L105 and Fig.1). It is thus not possible to conclude the role of groundwater in contributing to the streamflow and the groundwater- surface water interactions. The subsurface lateral flow can be

high and not negligible in such large mountainous basin (>2*105 km2 ). It is recommended to be cautious in interpreting and concluding the result regarding the baseflow. All mentioning of groundwater baseflow in the manuscript actually refer to the subsurface flow, i.e. the sum of both unsaturated and saturated zone, e.g. on L134, it is subsurface flow, but not baseflow. The presented modeling approach is not able to investigate groundwater alone.

We thank the reviewer for this detailed and constructive comment. We agree that the representation of subsurface processes in our model is simplified, and we acknowledge that subsurface flow in the current setup comprises both lateral flow in the unsaturated zone (often referred to as interflow) and baseflow from the saturated zone. These two components are indeed treated as a single aggregated term in the model, and as such, it is not possible to explicitly separate or quantify the contribution of groundwater to streamflow, nor to analyze groundwater–surface water interactions in detail.

In light of the reviewer's observation, we will revise the manuscript to replace the term "baseflow" with "subsurface flow" throughout, when referring to the total contribution from both the unsaturated and saturated zones. This change will be made, for instance, in L134 and similar occurrences. We will also include a clarification early in the methods section to define explicitly what is meant by "subsurface flow" in the context of our model.

Moreover, we will address this structural simplification as a modeling limitation and clarify that the current modeling approach does not allow us to investigate groundwater dynamics in isolation. We fully agree that caution is required when interpreting model outputs in terms of baseflow or groundwater contributions, especially in a large mountainous basin where interflow can be substantial.

Finally, when referring to streamflow during dry periods, we propose using the term "low flow" to avoid potential misunderstandings and to emphasize that our analysis focuses on the overall behavior of the hydrograph during low-flow conditions, rather than on baseflow in the strict hydrogeological sense.

Regarding the modeling, are the spatial zones delineated the same for both the surface and subsurface? (this could potentially fragment the aquifers located at the boundaries). Is the subsurface flow allowed to cross the delineated boundaries? The conceptualization of the subsurface processes in the model potentially limits the ability of the model for investigating the surface-subsurface interactions. The model limitation should be clearly discussed in Section 4.3.

In the model, only the runoff concentration process through the river network is considered to occur crossing the boundaries of simulation units (i.e., REWs). The runoff generation processes (both surface and subsurface) only occur within the REW. The model actually only considers the shallow groundwater, which is frequently recharged by the infiltration, but does not consider the deep groundwater cycle. This is indeed a limitation in model, especially for the TP, where previous studies indicated deep interbasin groundwater pathways existed. We will discuss these in the limitation.

There are 4 discharge stations, but only the results at Nuxia Station are presented. The results for the other stations should be presented in the Supplementary Information.

The authors should also clarify if the conclusions achieved at Nuxia Station are held the same as the other three stations.

We thank the reviewer for pointing this out. Although there are four national discharge stations in the basin, the data are not publicly accessible and can only be requested from the provider, subject to approval and specific conditions. We were able to obtain data for only two additional stations—Yangcun and Nugesha—besides Nuxia. In the revised version of the manuscript, we will include the results for Yangcun and Nugesha in the Supplementary Information. We will also clarify that the conclusions drawn at Nuxia Station are consistent with those observed at these two additional stations, as shown in Figures 1 and 2.

Does glacier melt contribute to groundwater recharge? Or is it assumed that all glacier melt goes into streamflow? This assumption should be clear in the text as well.

Yes, we assume that glacier melt generates streamflow directly through the surface pathway, because of the low permeability of the glacier surface. We will clarify this in the revised manuscript.

The simple degree-day-factor methods are used to solve snowmelt and glacier melt. Glacier mass balance is estimated with a simple volume-area scaling factor approach. The limitations of these adopted simple approaches for solving snow and glacier melts should be discussed in terms of modeling limitations.

These methods are indeed rather simplified, and the adoption of a spatially uniform degree-day factor cannot reflect the spatial heterogeneity of the melting processes. We will discuss them in the limitation section. The reason why we adopted a simplified method is that the study basin is very large for a hydrological model, and we need to make the model adequately efficient for the subsequent GLUE analysis. Besides, although the degree-day factor method is simplified, it is a rather commonly used method, and performs effectively in snow and glacier simulation (especially when we are concerned about the characteristic at a large spatial scale).

L213: how are the Pareto fronts defined? Please justify this threshold used to show the Pareto fronts in Figure 3 and the conclusions obtained from this result relating to this threshold.

The Pareto fronts shown in Figure 3 (red points) are defined based on the standard multi-objective dominance criterion, where no other solution performs better across all objectives and strictly better in at least one. The blue dashed lines are not part of the Pareto front definition but are used to indicate performance thresholds (e.g., NSE > 0) that help distinguish between simulations with at least minimal predictive value and those that are clearly inadequate. In particular, an NSE value below zero indicates that the model performs worse than a simple baseline given by the mean of the observed data. The NSE = 0 line is therefore used as a permissive reference to highlight the broad performance landscape, including poor-performing solutions. A similar reasoning applies to the glacier mass balance indicator VE: values below zero indicate that the mean simulated mass balance deviates from the observed mean more than a null reference would (i.e., assuming zero glacier mass change), and are therefore also interpreted as indicative of non-behavioral model realizations. These thresholds support a transparent visualization of the trade-off space and enable a clear separation of solutions with meaningful predictive skill. The conclusions regarding trade-offs remain consistent across different thresholds, and the ones chosen here serve to support a meaningful

interpretation of solution clusters. We acknowledge that this methodological choice and its implications were not clearly explained in the initial version of the manuscript. To improve clarity, we will add a dedicated paragraph to explicitly justify the threshold selection and discuss its influence on the interpretation of the results.

L248-249: Why does the KKA shows a noticeable convergence, but not KKD? They both are parameters that control the subsurface runoff outflow rates. Please clarify this point.

We acknowledge the reviewer's comment and agree that the contrast in convergence between KKA and KKD deserves clarification. Although both parameters influence the subsurface runoff outflow, their functional roles in the model differ significantly. KKA is an exponential coefficient, meaning that even small changes in its value can lead to large nonlinear variations in the outflow rate. This makes KKA highly sensitive and more easily constrained by the calibration targets. On the other hand, KKD acts as a linear coefficient, whose effect on the runoff response is more gradual and can be compensated by other interacting parameters. As a result, KKD tends to be less identifiable, leading to a flatter posterior distribution and a lack of noticeable convergence. We will clarify this point in the revised manuscript.

**Data perspective:**

Limited information is provided on the input data of this study. This could hamper the readers to interpret the results.

L79: The four river gauging are only given by names and no other information and data are available. It is recommended to provide details on the coordinates and elevations of the four river gauging stations in this mountainous basin, also their observed periods, frequency, and measurement method. Any observations errors/failures in the winter low flow and high flow periods? These details are important to interpret the observed and simulated discharge.

We appreciate the suggestion and will include a table summarizing the basic information of the hydrological stations utilized in our study. However, it's important to note that, despite being part of China's national monitoring network, detailed information—such as specific measurement methodologies and potential errors during low or high flow conditions—is not publicly accessible. Our access to these data was facilitated through personal connections, reflecting the broader challenges associated with water data availability in China. This situation aligns with the issues highlighted by Lin et al. (2023), who discuss the limited accessibility and usability of China's water data and advocate for the development of a national water data infrastructure to enhance data sharing and support effective water resource management.

| Station | Coordinate | Elevation (m) | Streamflow | Isotope | | | | |
|---------|-----------|---------------|------------|---------|---|---|---|---|
| | | | Period | Period (in 2005) | Precipitation | | Stream water | |
| | | | | | Sample number | $\overline{\delta^{18}O}$ (‰) | Sample number | $\overline{\delta^{18}O}$ (‰) |
| Nuxia | 94.65°E, 29.47°N | 3691 | 2001-2015 | 14 Mar-23 Oct | 86 | -10.33 | 34 | -15.74 |
| Yangcun | 91.82°E, 29.27°N | 4541 | 2001-2010 | 17 Mar-5 Oct | 59 | -13.14 | 30 | -16.57 |
| Nugesha | 89.71°E, 29.32°N | 4715 | 2001-2010 | 14 May-22 Oct | 45 | -14.29 | 25 | -17.84 |
| Lazi | 87.58°E, 29.12°N | 4889 | / | 6 Jun-22 Sep | 42 | -17.41 | 22 | -16.52 |

Table 1: Data and sample information at four stations

Section 2.1: What is the modelling period? Please detail the start and end dates of the meteorological data sets and the modelling period. Also add details on which years of DEM, land use data, soil data, snow cover, and glacier data are used in this modelling study.

The modelling period is 2001.1.1-2015.12.31, which is also the start and end date of the meteorological data we have adopted.

Details of how each dataset used in the model:

- DEM is used for dividing the whole basin into several REWs, and calculating the REW attributions for model calculation (e.g., the basin area, the length and width of river channel, the slope of the hillslope).
- The vegetation data (NDVI and LAI) is used to determine the proportion of vegetation covered area (i.e., the vi-zone and vd-zone in the model) and the interception ability.
- The soil data is used to determine the soil properties not obtained by calibration, including saturated hydraulic conductivity, soil porosity, soil pore distribution index, field capacity, and air entry value, which are used for the simulations of infiltration, exfiltration and groundwater outflow processes.
- The snow cover data is used for the calibration of SCA simulation.
- The glacier cover area data is used for determination of the boundary of regions where glacier simulation is conducted, and for dividing the glacier simulation unit. The glacier elevation change data is used for the calibration of GMB simulation.

More detailed descriptions of each dataset and its role in the modeling framework are provided in the main text of the paper.

3. L79-93: Are the gridded meteorological satellite data corrected with in-situ station data? How are the different resolutions of various types of gridded spatial data used in the hydrological model? Please provide details on this.

The correction of satellite data with in-situ station data is not conducted in our study, but this process is included in the production process of some datasets, such as the precipitation and temperature data in the CMFD dataset (He et al., 2020: the first high-resolution meteorological forcing dataset for land process studies over China | Scientific Data). The only data correction conducted by ourselves is to correct the isoGSM precipitation isotope data by the measurement data, which will be explained in detail in the response to the next comment.

The simulation unit of the model is REW, the average area of which is ~700 km2, larger than the spatial resolution of all the gridded products. The areal averages of each factor are calculated in each REW, which are used as the input for the simulation.

We will explicitly state this in the revised version of the manuscript to improve clarity.

The description of the streamflow sampling is very vague, which is simply stated as "Grab samples of stream water were collected in 2005 at four stations..". Please provide details on how many samples and in which months the samples were collected. Do the authors have the precipitation (rainfall, snow) isotopes in the same year (2005) or in a different year (2008)? Using streamflow and precipitation isotope data of different years in the same model can be inappropriate.

We will provide a table to show the detailed information of the precipitation and river water samples, including the number of samples, the sampling period, and the isotopic characteristics. Precipitation samples were also collected in 2005 for the same period. The isotope data of these precipitation samples were used to correct the gridded output of the isotopic general circulation model isoGSM, to drive our model. We will add a description of this process in the data section of the revised manuscript.

How is the precipitation tracer estimated for rainfall and snow individually? This needs to be clarified in the manuscript.

As explained in the last response, the precipitation isotope produced by the isoGSM corrected based on observation precipitation isotope data was adopted as the model input. The isotope compositions of rainfall and snowfall were assumed to be the same, but that of snowpack and snowmelt in the catchment was simulated based on the balance equations of water and isotope mass, similarly as other water bodies.

We will clarify these aspects in the revised manuscript.

**Interpretation of the results:**

L256-259: The SCA shows a higher influence on the posterior distribution of T0 than the GMB, which does not show the strongest influence as the authors interpreted. Could the authors please clarify why they see GMB as the strongest from this result figure (Fig.4j)?

Thank you for pointing this out. You are absolutely right: the SCA shows a stronger influence on the posterior distribution of $T_0$ compared to the GMB, as is clearly evident in Figure 4j. We acknowledge this misinterpretation in our original statement and will correct the text accordingly in the revised manuscript to accurately reflect the results.

L306-308: The isotope data have increased uncertainty of the simulated glacier melt runoff (Fig.6d), but they are helpful to constrain other surface runoff components (rainfall runoff, snowmelt). Please clarify this result.

Glacier meltwater is assumed to generate runoff directly through surface pathway, so its contribution is not related to the partitioning between surface and subsurface runoff, for which isotope is helpful. Consequently, isotope cannot reduce the uncertainty of glacier meltwater.

L272-277: Including the isotope data leads to a decreased containing ratio. This means a significant under capture of the extremely low and high streamflow. Why including isotope data has decreased the streamflow simulation performance? Please clarify this result.

We thank the reviewer for this question. The observed reduction in the containing ratio when including isotope data is a consequence of the additional constraints introduced by the isotopic information. Isotope observations provide insights into flow paths and residence times, which reduce the set of parameter combinations that are simultaneously consistent with both hydrological and isotopic signatures. This leads to a more selective model response, where some simulations that were previously acceptable based on streamflow alone are now rejected due to inconsistencies with the isotope data. This effect is not limited to extreme flows, but applies more generally to the full range of simulated dynamics. It highlights the trade-offs that emerge in multi-objective calibration, where integrating multiple sources of information typically narrows the behavioral parameter space. The reduction in the containing ratio is particularly marked when compared to other observational targets such as snow cover area or glacier mass balance. While these variables mainly constrain the water balance and storage dynamics, isotope data provide independent information on flow partitioning and subsurface mixing processes. This imposes stronger internal constraints on the model structure and functioning, making it harder to compensate structural mismatches through parameter adjustment alone. As a result, fewer parameter sets are able to simultaneously satisfy both streamflow and isotope-based objectives. This highlights the added value of isotope data for model diagnosis and internal consistency, but also explains the greater selectivity they introduce during calibration.

**Technical corrections:**

L8: It would be helpful to mention which type of hydrological model the THREW-T is in the abstract. i.e. fully-distributed, semi-distributed, or conceptual?

L78: km2 should be straight upright, not italic. Please correct all formats of the units for similar cases.

L80, L82: Please add years between which the mean annual precipitation and mean annual temperature are calculated.

L100: distributed -> semi-distributed?

L104: what is bare zone? Bare soil, bare rock?

Figures 2 and 4: avoid using red and green colors together in the same figure to allow readers with colour vision deficiencies to correctly interpret your findings.

Table 1: table caption should be on top of the table.

L165: Please correct the formats of Equations 1-6 by following the journal guideline. e.g. the NSE should be straight upright, not italic. The text subscription should be straight upright as well.

L210-211: NSE, VE should be straight upright, not italic. Please check the format of all such mentioning.

L266-267, 281: Please remove the parentheses around the Section and Figure numbers, and correct all such mentioning in the manuscript.

Figure 4 caption i) covered area -> snow covered area.

Thank you for your careful review and helpful suggestions. We will address all the technical corrections you pointed out in the revised version of the manuscript.

[Figure]

*Fig. 1: The 5–95% percentile prior, conditioned solely on streamflow, and posterior predictive uncertainty ranges for streamflow, calculated under different conditions: snow cover area (SCA), glacier mass balance (GMB), and isotopes (I) at Yangcun gauging station . Left panels: daily streamflow time series for the period 2005–2010; right panels: flow duration curves for the entire period 2001–2015.*

[Figure]

*Fig. 2: The 5–95% percentile prior, conditioned solely on streamflow, and posterior predictive uncertainty ranges for streamflow, calculated under different conditions: snow cover area (SCA), glacier mass balance (GMB), and isotopes (I) at Nugesha gauging station . Left panels: daily streamflow time series for the period 2005–2010; right panels: flow duration curves for the entire period 2001–2015.*

**Bibliography:**

He, J., Yang, K., Tang, W. et al. *The first high-resolution meteorological forcing dataset for land process studies over China.* Sci Data 7, 25 (2020). https://doi.org/10.1038/s41597-020-0369-y

Lin, J., Bryan, B.A., Zhou, X. *et al.* Making China's water data accessible, usable and shareable. *Nat Water* **1**, 328–335 (2023). https://doi.org/10.1038/s44221-023-00039-y

Nan, Y., & Tian, F. (2024). *Glaciers determine the sensitivity of hydrological processes to perturbed climate in a large mountainous basin on the Tibetan Plateau.* Hydrology and Earth System Sciences, 28(3), 669–689. https://doi.org/10.5194/hess-28-669-2024

Nan, Y., He, Z., Tian, F., Wei, Z., & Tian, L. (2021). *Can we use precipitation isotope outputs of isotopic general circulation models to improve hydrological modeling in large mountainous catchments on the Tibetan Plateau?* Hydrology and Earth System Sciences, 25(12), 6151–6172. https://doi.org/10.5194/hess-25-6151-2021

---

## Author Response (AR1)

**Dear Editor,**

We thank you for handling our manuscript and we appreciate the effort that Referees have put into their assessment. Please enclosed you can find the revised version of the manuscript titled "Reducing Hydrological Uncertainty in Large Mountainous Basins: The Role of Isotope, Snow Cover, and Glacier Dynamics in Capturing Streamflow Seasonality", reference number EGUSPHERE-2025-664.

After having carefully read their comments, we believe we fully addressed each point as reported in the attached rebuttal document. Please also find enclosed a pdf document which details in track changes mode all the revisions we included into the revised manuscript.

We do believe that the revised manuscript improved significantly and meets the quality standards of the HESS journal.

Diego Avesani on behalf of the authors

Address

Department of Civil, Environmental and Mechanical Engineering, University of Trento, via Mesiano 77, 38123 Trento, Italy email: diego.avesani@unitn.it

**Reply to Editor and Reviewers**

We thank the Editor and the Referees for the valuable comments. Below we reply point to point and describe the modifications introduced in the revised version of the manuscript. Our replies are evidenced in green.

**Reply to Editor**

Dear authors,

As you have seen, two reviewers have provided excellent, constructive and very detailed comments on your manuscript. They both, overall, appreciate your analysis and think that it can be a very valuable contribution to literature. However, they both also flag a number of critical issues that need to be resolved. I largely agree with that assessment.

From my perspective, the two most relevant points arising are the following:

- (1) the choice to limit the distinction of water pathways to only two components is rather simplistic and may lead to misinterpretation of the results. Both reviewers have provided alternative approaches that can provide a bit more process detail and that may eventually strengthen the overall findings of your analysis. It will thus be a good idea to heed the reviewers advice and explore different options to test whether more information about the hydrological functioning can be obtained from defining more endmembers.
- (2) although you have replied to the reviewer concern about the data availability, the data policy of HESS is unambiguous: "If the data are not publicly accessible at the time of final publication, the data statement should describe where and when they will appear, and provide information on how readers can obtain the data until then. Nevertheless, authors should make such embargoed data available to reviewers during the review process in order to foster reproducibility. The Copernicus review system allows to define such assets as 'access limited to reviewers' and reviewers must then sign that they will use such data only for the purpose of reviewing without making copies, sharing, or reusing. In rare cases where the data cannot be deposited publicly (e.g., because of commercial constraints), a detailed explanation of why this is the case is required. "(https://www.hydrology-and-earth-system-sciences.net/policies/data\_policy.html). I thus politely request you to follow this policy and to add the required information in the revised version of the manuscript.

Once you have addressed and incorporated these and all other reviewer comments, I am looking forward to receiving a revised version of your manuscript.

Best regards, Markus Hrachowitz

**Reply**

We thank the Editor for his assessment and for the opportunity to submit a revised version of the manuscript. We took in great considerations all Referees' comments and in the revised manuscript we introduced the following modifications:

• Regarding the data availability issue, we fully acknowledge the data policy of HESS and are committed to complying with it. As the Yarlung Tsangpo River is a transboundary river, streamflow measurement data are classified as nationally confidential by Chinese authorities and cannot be publicly released. This constraint is also emphasized in a recent perspective article (Lin et al., 2024), which highlights the particular sensitivity of water data in transboundary river basins and regions affected by geopolitical tensions. Consequently, we had to obscure the y-axis in figures that display observed streamflow. Nevertheless, in line with the journal's policy, we have made the simulated streamflow data openly available via Zenodo, and have provided the corresponding link in the revised Data Availability section. We believe that sharing the simulation outputs – together with a clear statement in the manuscript explaining the data limitations – offers transparency and supports reproducibility to the extent permitted by national regulations.

\_\_\_\_\_

**Reply to Review 1:**

The manuscript presents a hydrological modeling study in a glacier-influenced catchment. The work explores the value of auxiliary datasets, namely water isotope composition, snow cover area, and glacier mass balance in model calibration in a GLUE framework. The model structure allows tracer simulations and comparison with spatially variable datasets. The works finds different datasets have more power in model calibration in different hydrological seasons: isotopes during baseflow, and snow and glacier related observations during the melt period.

I liked the systematic approach for including model validation datasets of very different origin to model evaluation scheme. The GLUE uncertainty analysis framework for the work is in my judgement valid. The overall approach the authors develop to explore parameter sensitivity to model validation objectives and stream source water contribution are in my opinion of interest to the community. I recommend the work to be published after addressing my comments below:

**Reply**

We thank the Referee for the overall positive assessment of our study and the encouraging comment.

**MAJOR COMMENTS**

I'd like to see better presentation of the stable water isotope data. You have only isotope data of the streamflow validation, it remains unclear how representative the input precipitation data is of the catchment. Do any of the references cited for the model development have any comparison data for simulated precipitation, snow or groundwater isotope composition? Having even cursory validation of the simulated isotope composition in different model compartments (snow, glacial melt, groundwater mainly) in the would give more credibility that the streamflow isotopes are correctly simulated and informative for the right reasons. On that note, I'd like to see a figure of the stream isotope data and model simulation fit to stream isotopes.

**Reply**

We thank the reviewer for raising this point. We have added more detailed descriptions of the isotope characteristics for various water bodies, see lines 103-110 of the revised manuscript. Specifically:

• Precipitation: We now clarify that our previous evaluation of isoGSM (Nan et al., 2021) showed it can reasonably capture the seasonal variation in

precipitation  $\delta^{18}$ O, though it tends to overestimate values and struggles with event-scale variability (see Figures S1 and S2 of supplementary material). To address this, we used a corrected isoGSM product developed in Nan et al. (2022), which adjusts the original isoGSM values through a regression-based bias correction with altitude. Importantly, this corrected product assimilates observed  $\delta^{18}$ O data when available, so it is not appropriate to compare it directly against observations; we therefore present comparisons only between the original isoGSM and the observed data, to illustrate the model's raw performance.

- Glacier melt: We now explain that glacier melt  $\delta^{18}$ O was estimated using the offset-parameter method, assuming a constant value 5‰ lower than the altitude-weighted average of local precipitation  $\delta^{18}$ O. This offset was based on data from Boral and Sen (2020), and the value is supported by previous studies in similar environments.
- Groundwater: While groundwater samples were not available for isotope validation, we discuss that groundwater  $\delta^{18}O$  typically shows low temporal variability compared to precipitation or streamflow, due to the long residence time. This characteristic has been included in our discussion of isotope contributions.
- Snowmelt: We were also unable to collect snowmelt  $\delta^{18}O$  samples. Consequently, isotope likelihoods did not significantly constrain snow simulations or snowmelt runoff estimates. These remain primarily informed by snow cover area data, as described in the revised methods section.
- Streamflow: We have added a figure (now Figure~S5 in the Supplementary Material) that compares the observed and simulated  $\delta^{18}O$  in streamflow, to demonstrate the model's capability in reproducing isotopic dynamics.

The fractions for snowmelt surface runoff and glacier surface runoff seem low to me. Can you provide comparison with fractions found in other montanous snow and glacier influenced sites? Quite often end-member mixing analysis fraction estimations are done for three end members: snow, rain and glacial melt. In your model analysis groundwater is explicitly considered as a component, but isotopically it is essentially composed of rain, snow and glacial melt. This in my opinion creates a bit of confusion, and makes the glacial and snow melt seem less important for the regions water resources. I don't think there is an error in your analysis, but would be good to clarify the concepts further, to make your results more relatable to other literature.

**Reply**

We thank the reviewer for raising the important point regarding the definition of runoff components. In the revised manuscript, we have added a discussion to clarify the rationale behind our choice and how it compares with previous studies.

Specifically, in lines 146–150 of the revised version, we now explain that two common approaches exist to define runoff components: (1) by source (rainfall, snowmelt, glacier melt) and (2) by generation pathway (surface vs. subsurface runoff). Since groundwater is also important in the region, reporting results for both sets of definitions would require parallel accounting (e.g., rainfall 80%, snowmelt 10%, glacier melt 10%; surface runoff 40%, subsurface runoff 60%), which could cause confusion. Therefore, we adopted a hybrid approach and defined four components to provide a more informative and concise framework.

We also added that the estimated contribution of each component depends strongly on the definition adopted and the datasets used for model validation. The relatively low contribution of snowmelt and glacier melt in our results is partly due to the inclusion of a large share of subsurface runoff. However, our estimates are consistent with other studies that use snow and glacier data for model validation. For example, Chen et al. (2017) reported contributions of 10.6% and 9.9% for snowmelt and glacier melt, respectively, while Zhang et al. (2025) reported 6.0% and 6.2% using different normalization schemes. When adopting similar definitions, our estimates closely align with theirs, reinforcing the robustness of our results.

**MINOR COMMENTS**

L12: I perceive GW-SW interactions as specific water exchange processes between surface and subsurface water. As you don't really delve deeper into GW-SW interactions in your simulations, I'd propose that you stick with talking only about baseflow, not GW-SW interactions (which baseflow generation if of course a manifestation of)

**Reply**

Thank you for this helpful clarification. We agree that the term "groundwater—surface water interactions" may be misleading in the context of our study, as our model does not explicitly simulate these processes. In the revised manuscript, we have removed this term and now refer more appropriately to "subsurface flow," which better reflects the structure of our model and is consistent with the scope of our analysis. This revision also aligns with the observation raised by Reviewer 2.

L54-L66: seems like the research questions are to some extent repeated. Suggest to review and rewrite more concisely.

**Reply**

Thank you for the suggestion. We acknowledge the redundancy in the original paragraph and have revised it in the updated manuscript to make the research questions more concise and clearly focused.

L110: Do you think snow sublimation would be a significant flux in your region, possibly influencing the snow storage and isotope composition of the snowpack consequently snow melt?

**Reply**

Thank you for raising this point. Some studies have indicated that, due to the relatively wet conditions in the YTR basin, sublimation losses are minor, typically accounting for only 2–3% of annual snowfall (Lutz et al., 2016; Khanal et al., 2021). Accordingly, and consistent with other modeling studies in the region (e.g., Chen et al., 2017; Sun et al., 2024), we did not include snow sublimation processes in our model. This assumption has now been clarified in the revised manuscript at lines 125-128.

L213: not clear how the simulations comprising the pareto front (red markers in are selected. seems like the number of the included simulations is fairly low, around 15.

**Reply**

Thank you for your observation. The red markers in the figure represent simulations that lie on the Pareto front, identified in the bi-objective space illustrated in each panel. The relatively small number of these points (~15) is due to the fact that only a limited subset of simulations are non-dominated with respect to both objectives. We would like to clarify that the Pareto front is computed across the entire simulation ensemble and is not influenced by any behavioral classification criteria. Therefore, the red points should not be interpreted as behavioral simulations but rather as Pareto-optimal solutions based solely on the two plotted performance metrics. This is consistent with the findings of Di Marco et al. (2021), who also reported that the number of Pareto-optimal simulations is often substantially lower than that of behavioral ones, underscoring how multi-objective trade-offs can yield highly selective solution subsets. We have revised the figure caption and the corresponding text in the manuscript to clarify this distinction (lines 229-243 in the revised version).

L238: can you further explain where the prior parameter distributions in Fig.4 comes from. Is it the parameters with >0 NSE for streamflow?

Reply

We agree that the origin of the prior parameter distributions shown in Fig. 4 requires further clarification. The prior parameter distributions are derived from model parameter sets that resulted in a positive Nash–Sutcliffe Efficiency (NSE>0) for streamflow. This filtering step ensures that only behaviorally plausible parameterizations are included in the prior.

We have revised the main text to explicitly describe this criterion to improve clarity (lines 264–271 in the revised manuscript).

**L304: I don't fully understand why the sensitive LL parameter does not manifest in the snowmelt fraction.**

**Reply**

Thank you for this insightful comment. From a water balance perspective, the contribution of snowmelt is primarily governed by the fraction of snowfall in total precipitation, which depends on the temperature threshold used for rainfall/snowfall partitioning. While the LL parameter mainly affects the spatial extent of snow cover — and thus the spatial and temporal distribution of snowmelt — its influence on the total amount of snowmelt remains limited. We have clarified this point in the revised manuscript (lines 274–277).

L307: the narrower ranges for isotope simulations are not evident visually compared to the Q simulations. Would any statistical test either looking for differences in central values or variability in the distributions be helpful in identifying the differences?

**Reply**

We thank the reviewer for this insightful suggestion. We considered conducting statistical tests to assess differences in central tendency or dispersion between the posterior distributions. However, we found that emphasizing these differences visually was more effective in this context. To this end, we added an inset panel in Figure 5, which allows for a clearer comparison between the distributions. The quantification and interpretation of these differences are discussed in the revised Discussion section, where both the sharpness and the containment ratio are used as metrics to characterize the differences between the prior and posterior distributions.

**L310: incomplete sentence?**

**Reply**

Thank you for pointing this out. We corrected the manuscript accordingly.

L341-343: not very clear how successful the snow cover extent simulations are in the first place. The NSE metric is not very intuitive for snow cover extent variable. If for example the extent in area does not quantify, if the snow cover is simulated in the correct location. Similarly as requested for the isotopes, can you provide the timeseries of observed vs simulated snow cover extent to identify and discuss some potential some biases.

**Reply**

Thank you for this comment. We agree that NSE, while commonly used, may not fully capture spatial characteristics of snow cover dynamics. In our analysis, we focus on the catchment-integrated snow-covered area (SCA), for which NSE remains a useful metric to evaluate the agreement between observed and simulated temporal patterns of areal extent. To better illustrate model performance, we have included the time series of observed versus simulated SCA in Figure S3 of the Supplementary Material, along with analogous comparisons for glacier mass balance (GMB) and isotopic signatures in Figures S4 and S5. These visualizations allow the reader to assess the temporal evolution and potential systematic biases for each variable. As described in Section 3.4, and detailed in lines 375–381, of the revised manuscript, the figures also display the corresponding posterior predictive uncertainty ranges.

L248-249: Why does the KKA shows a noticeable convergence, but not KKD? They both are parameters that control the subsurface runoff outflow rates. Please clarify this point.

**Reply**

We thank the reviewer for this useful observation. We agree that the contrasting convergence behavior of parameters KKA and KKD warrants clarification. Although both parameters affect subsurface runoff outflow, their functional roles in the model differ. KKA is an exponential coefficient, and even minor variations in its value can produce strong nonlinear changes in the simulated outflow. This sensitivity makes KKA more responsive to the calibration constraints, resulting in a sharper posterior distribution. Conversely, KKD is a linear coefficient, whose impact on runoff is more gradual and can often be offset by compensatory effects from other parameters. This structural compensation reduces identifiability, leading to a flatter posterior and limited convergence. We have clarified this explanation in the revised manuscript (lines 282–287).

**Data perspective:**

Limited information is provided on the input data of this study. This could hamper the readers to interpret the results.

L79: The four river gauging are only given by names and no other information and data are available. It is recommended to provide details on the coordinates and elevations of the four river gauging stations in this mountainous basin, also their observed periods, frequency, and measurement method. Any observations

errors/failures in the winter low flow and high flow periods? These details are important to interpret the observed and simulated discharge.

**Replay**

We appreciate the reviewer's suggestion and have now included a table summarizing the basic information of the hydrological stations used in our study. However, we wish to note that, although these stations are part of China's national hydrological monitoring network, detailed metadata—such as measurement protocols or error characterizations under extreme flow conditions—are not publicly accessible. Our access to the discharge data was made possible through personal connections, which reflects broader challenges in water data availability across China. This limitation is consistent with the issues reported by Lin et al. (2023), who emphasized the restricted accessibility and usability of hydrological data in China and called for the development of a national water data infrastructure. These clarifications have been added to the revised manuscript (lines 95–97).
* * *
**Reply to Review 2:**

**General comments:**

This manuscript focuses on evaluating the value of snow cover area, glacier mass balance, and isotopes in reducing uncertainty and equifinality of hydrological modeling in a large mountainous basin in the Tibetan Plateau. The Bayesian approach and GLUE method are adopted to investigate the research questions. The research topic aligns with the journal scope and the research findings are potentially useful for the readers. I have a few concerns regarding the modeling procedure, the details of the input data, and the interpretation of the results before the paper being accepted for publication.

Additionally, one thing I noticed here is that the time-series simulated and observed discharge does not have a y-axis (Fig.5), which is present on purpose due to data dissemination restrictions mentioned in the caption. However, this is not possible for readers to understand the model performance, and the magnitude of the simulated and observed discharge. A manuscript avoiding showing y-axis of time-series discharge plot in the results could potentially conflict with the basic principle of open science of HESS/Copernicus journals.

**Specific comments:**

**Modeling perspective:**

The subsurface is overly-simplified represented in the model. The subsurface flow generates from the model is composed of the subsurface lateral flow ("interflow") in the unsaturated zone and the baseflow from groundwater to surface water in the saturated zone. These two subsurface flow components are simulated as a sum (L105 and Fig.1). It is thus not possible to conclude the role of groundwater in contributing to the streamflow and the groundwater- surface water interactions. The subsurface lateral flow can be high and not negligible in such large mountainous basin (>2\*105 km2). It is recommended to be cautious in interpreting and concluding the result regarding the baseflow. All mentioning of groundwater baseflow in the manuscript actually refer to the subsurface flow, i.e. the sum of both unsaturated and saturated zone, e.g. on L134, it is subsurface flow, but not baseflow. The presented modeling approach is not able to investigate groundwater alone.

**Replay**

We thank the reviewer for this detailed and constructive comment. In response, we revised the manuscript to clarify that in our modeling framework, subsurface flow comprises both lateral flow in the unsaturated zone (commonly referred to as interflow) and baseflow from the saturated zone. As these two components are treated as a single aggregated term, it is not possible to explicitly quantify

groundwater contributions or investigate groundwater-surface water interactions.

Accordingly, we have replaced the term baseflow with subsurface flow throughout the manuscript (e.g., line 134 and similar occurrences), to ensure terminological consistency. A clarification of this definition has also been added in the Methods section. Furthermore, we now explicitly acknowledge this structural simplification as a model limitation, and we caution against interpreting our results in terms of baseflow or groundwater dynamics. Lastly, in places where streamflow during dry periods is discussed, we replaced baseflow with low flow, to emphasize that we are referring to overall hydrograph behavior, rather than to baseflow in the strict hydrogeological sense.

Regarding the modeling, are the spatial zones delineated the same for both the surface and subsurface? (this could potentially fragment the aquifers located at the boundaries). Is the subsurface flow allowed to cross the delineated boundaries? The conceptualization of the subsurface processes in the model potentially limits the ability of the model for investigating the surface-subsurface interactions. The model limitation should be clearly discussed in Section 4.3

**Replay**

We thank the reviewer for raising this important point. In the model, only the runoff concentration process through the river network is allowed to cross the boundaries of the simulation units (i.e., Representative Elementary Watersheds, REWs), while runoff generation – both surface and subsurface – occurs entirely within each REW. The model accounts only for shallow groundwater, which is frequently recharged by infiltration, and does not simulate the deeper groundwater cycle. We acknowledge this as a structural limitation of the model, particularly in the context of the Tibetan Plateau, where previous studies have highlighted the existence of deep interbasin groundwater pathways. This limitation is now explicitly discussed in the revised manuscript (lines 480–485).

There are 4 discharge stations, but only the results at Nuxia Station are presented. The results for the other stations should be presented in the Supplementary Information. The authors should also clarify if the conclusions achieved at Nuxia Station are held the same as the other three stations.

**Replay**

We thank the reviewer for pointing this out. Although four national discharge stations exist in the basin, the data are not publicly accessible and can only be obtained upon request, subject to approval and specific conditions. We were able to acquire discharge records for only two additional stations, Yangcun and Nugesha, in addition to Nuxia. In the revised manuscript, we now include the corresponding results in the Supplementary Information. As shown in Figures

S6 and S7, and explicitly stated in the main text (lines 328–331), we clarify that the conclusions drawn at Nuxia are consistent with those observed at Yangcun and Nugesha, thereby strengthening the robustness of our findings.

Does glacier melt contribute to groundwater recharge? Or is it assumed that all glacier melt goes into streamflow? This assumption should be clear in the text as well.

**Replay**

We thank the reviewer for this comment. Yes, in our model setup, we assume that glacier melt contributes directly to streamflow via the surface runoff pathway due to the low permeability of glacier surfaces. This modeling assumption has now been explicitly clarified in the revised manuscript (see lines 134–137).

The simple degree-day-factor methods are used to solve snowmelt and glacier melt. Glacier mass balance is estimated with a simple volume-area scaling factor approach. The limitations of these adopted simple approaches for solving snow and glacier melts should be discussed in terms of modeling limitations.

**Replay**

We thank the reviewer for raising this important point. We agree that the adopted methods are simplified and that using a spatially uniform degree-day factor does not capture the heterogeneity of melt processes across the basin. In the revised manuscript, we now explicitly acknowledge this limitation and explain our rationale for the chosen approach (lines 131–132). Specifically, due to the large spatial extent of the study basin and the need for computational efficiency in the subsequent GLUE analysis, we adopted the degree-day factor method, which, despite its simplicity, is widely used and has proven effective for snow and glacier simulations, particularly at large spatial scales.

L213: how are the Pareto fronts defined? Please justify this threshold used to show the Pareto fronts in Figure 3 and the conclusions obtained from this result relating to this threshold.

**Replay**

We thank the reviewer for this important observation. The red points shown in Figure 3 correspond to Pareto-optimal solutions defined using the standard dominance criterion: no other solution performs better across all objectives and strictly better in at least one. The blue dashed lines are not part of the Pareto front itself but are included to indicate minimal performance thresholds (e.g., NSE > 0) that help distinguish solutions with some predictive skill from those that are clearly inadequate. Specifically, NSE < 0 indicates that a model performs worse than the mean of the observations, and VE < 0 suggests a glacier mass balance deviation worse than a null model.

We acknowledge that the rationale for choosing these thresholds was not clearly articulated in the original version. In the revised manuscript (lines 256–262), we have now added a dedicated paragraph to explain the purpose of these thresholds and clarify that they are used for interpretative purposes, to better visualize the trade-off space, without altering the actual definition of the Pareto front. We also confirm that the conclusions regarding trade-offs are robust to different threshold selections.

Section 2.1: What is the modelling period? Please detail the start and end dates of the meteorological data sets and the modelling period. Also add details on which years of DEM, land use data, soil data, snow cover, and glacier data are used in this modelling study.

**Replay**

We thank the reviewer for the request to clarify the usage of datasets in our modeling framework. We have now added a dedicated paragraph in the revised manuscript (line 98) that provides a more concise and structured description of the role of each dataset and its relevance within the modeling period (2001–2015).

L79-93: Are the gridded meteorological satellite data corrected with in-situ station data? How are the different resolutions of various types of gridded spatial data used in the hydrological model? Please provide details on this.

**Replay**

We thank the reviewer for the helpful comment. We clarify that no additional correction of satellite data using in-situ station observations was performed in our study. However, some of the datasets we use — such as precipitation and temperature from CMFD — already include such corrections as part of their original data processing (e.g., He et al., 2020). The only correction we performed ourselves was for the precipitation isotope data from isoGSM, which we adjusted based on station measurements; this is explained in detail in response to the following comment. These clarifications have been added to the revised manuscript (lines 103–110).

Regarding spatial resolution, we note that the model operates at the scale of representative elementary watersheds (REWs), with an average area of approximately 700 km², which is larger than the resolution of the gridded datasets used. All input data are aggregated to the REW level as areal averages prior to simulation. These clarifications have been added to the revised manuscript (lines 118-120).

The description of the streamflow sampling is very vague, which is simply stated as "Grab samples of stream water were collected in 2005 at four stations..". Please provide details on how many samples and in which months the samples

were collected. Do the authors have the precipitation (rainfall, snow) isotopes in the same year (2005) or in a different year (2008)? Using streamflow and precipitation isotope data of different years in the same model can be inappropriate.

**Replay**

We thank the reviewer for this helpful comment. In the revised manuscript, we will provide a table summarizing the details of the precipitation and stream water samples, including the number of samples, sampling periods, and their isotopic characteristics. We confirm that precipitation samples were also collected in 2005, during the same period as the stream water samples. These precipitation isotope data were used to correct the gridded outputs of the isoGSM model, which serve as inputs to our hydrological model. A detailed description of this correction procedure has been added in the Data section of the revised manuscript (see lines 103–110).

How is the precipitation tracer estimated for rainfall and snow individually? This needs to be clarified in the manuscript.

**Replay**

As explained in the previous response, the precipitation isotope input used in our model was obtained from isoGSM outputs, which were corrected using observed precipitation isotope data. In the model, the isotopic compositions of rainfall and snowfall were assumed to be the same. However, the isotope composition of snowpack and snowmelt was dynamically simulated using mass balance equations for both water and isotopes, consistent with the treatment of other hydrological stores in the model.

**Interpretation of the results:**

L256-259: The SCA shows a higher influence on the posterior distribution of T0 than the GMB, which does not show the strongest influence as the authors interpreted. Could the authors please clarify why they see GMB as the strongest from this result figure (Fig.4j)?

**Replay**

Thank you for pointing this out. You are absolutely right: the snow-covered area (SCA) exerts a stronger influence on the posterior distribution of T0 compared to the glacier mass balance (GMB), as clearly shown in Figure 4j. We acknowledge the misinterpretation in our original statement.

L306-308: The isotope data have increased uncertainty of the simulated glacier melt runoff (Fig.6d), but they are helpful to constrain other surface runoff components (rainfall runoff, snowmelt). Please clarify this result.

**Replay**

We thank the reviewer for this observation. Glacier meltwater is assumed to generate runoff directly through the surface pathway in our model setup. Therefore, its contribution does not involve any partitioning between surface and subsurface components, the aspect for which isotopic data are most informative. As a result, the isotope likelihood does not help constrain the glacier melt contribution, leading to limited or no reduction in its associated uncertainty. This clarification has now been added to the revised manuscript (lines 459-467).

L272-277: Including the isotope data leads to a decreased containing ratio. This means a significant under capture of the extremely low and high streamflow. Why including isotope data has decreased the streamflow simulation performance? Please clarify this result.

**Replay**

We thank the reviewer for this insightful comment. The observed reduction in the containing ratio (CR) when including isotope data stems from the stronger constraints introduced by isotopic information. Isotopes provide orthogonal insights into flow partitioning and residence times, thereby narrowing the set of parameter combinations that are consistent with both streamflow and isotopic observations. This results in a sharper ensemble, where the spread of simulations is reduced and predictive confidence increases. However, this enhanced sharpness also increases the risk that observed streamflow values fall outside the uncertainty bounds, thus lowering the CR. Compared to other observational targets such as snow cover area (SCA) or glacier mass balance (GMB), which mainly constrain the seasonal water balance and storage dynamics, isotope data exert a stronger influence on internal hydrological processes. This leads to a more selective posterior and a reduced behavioral parameter space, highlighting the trade-off between sharpness and coverage (Beven et Binley, Gneiting et.al., 2007). While this underscores the diagnostic value of isotopic data in improving model consistency, it also suggests that further model improvements may be needed to achieve both sharpness and reliability. This explanation has been incorporated into the revised manuscript at lines 414-423.

**Technical corrections:**

- L8: It would be helpful to mention which type of hydrological model the THREW-T is in the abstract. i.e. fully-distributed, semi-distributed, or conceptual?
- L78: km2 should be straight upright, not italic. Please correct all formats of the units for similar cases.
- L80, L82: Please add years between which the mean annual precipitation and mean annual temperature are calculated.
- L100: distributed -> semi-distributed?

- L104: what is bare zone? Bare soil, bare rock?
- Figures 2 and 4: avoid using red and green colors together in the same figure to allow readers with colour vision deficiencies to correctly interpret your findings.
- Table 1: table caption should be on top of the table.
- L165: Please correct the formats of Equations 1-6 by following the journal guideline. e.g. the NSE should be straight upright, not italic. The text subscription should be straight upright as well.
- L210-211: NSE, VE should be straight upright, not italic. Please check the format of all such mentioning.
- L266-267, 281: Please remove the parentheses around the Section and Figure numbers, and correct all such mentioning in the manuscript.
- Figure 4 caption i) covered area -> snow covered area.

**Replay**

We thank the reviewer for these helpful technical corrections. We have carefully revised the manuscript and addressed all the suggested issues.

---

## Author Response (AR2)

Dear Editor,

We thank you for handling our manuscript, and we sincerely appreciate the time and effort that the Referee has dedicated to their thoughtful assessment.

Please find enclosed the revised version of the manuscript titled "Reducing Hydrological Uncertainty in Large Mountainous Basins: The Role of Isotope, Snow Cover, and Glacier Dynamics in Capturing Streamflow Seasonality", reference number EGUSPHERE-2025-664.

After carefully reviewing the Referee's comments, we believe we have fully addressed each point, as detailed in the attached rebuttal document. We also include a PDF version of the revised manuscript with all modifications clearly marked using track changes.

We believe that the manuscript has significantly improved and now meets the quality standards of Hydrology and Earth System Sciences. We have also ensured that the revised version **complies with HESS guidelines on data sharing and reproducibility**. In particular, we have updated the public data archive and clarified in the manuscript. We also believe that all shared elements are sufficient to support independent assessment and interpretation of the results, even within the constraints imposed by data confidentiality.

Sincerely,
Diego Avesani
on behalf of the authors

Address

Department of Civil, Environmental and Mechanical Engineering, University of Trento, via Mesiano 77, 38123 Trento, Italy email: diego.avesani@unitn.it

**Reply to Editor and Reviewers**

We thank the Editor and the Referee for the valuable comments. Below we reply point to point and describe the modifications introduced in the revised version of the manuscript. Our replies are evidenced in blue.

**Reply to Editor**

The reviewer is in principle satisfied with the proposed changes in the revised manuscript. There, however, remain a few open questions which I encourage you to address in the necessary detail. Please also make sure to give clear and detailed explanations of the data availability issue and a full description of the reasons what cannot be published and why. To do so follow the open data regulations provided on the HESS website.

**Reply**

As outlined in Section 2.1, streamflow data for the Yarlung Tsangpo River, which is part of a transboundary river system with China located upstream, are classified as confidential under Chinese national regulations. As a result, these data cannot be publicly disclosed, shared online, or included in any form of publication. This restriction reflects broader geopolitical considerations, as highlighted by Lin et al. (2023), who emphasize the heightened sensitivity surrounding hydrological data in transboundary basins, particularly in regions affected by resource-related or political tensions. This limitation has been explicitly acknowledged and discussed in the main text of the paper.

Such restrictions are not uncommon and have been acknowledged in several articles published in Hydrology and Earth System Sciences (HESS), where authors have transparently reported data confidentiality and addressed it through alternative data representations and detailed methodological documentation (e.g., Singh et al., 2023; Zhang et al., 2024). In line with HESS open data regulations, this study maintains scientific integrity by ensuring that all shared elements are sufficient to fully reproduce the results.

Nevertheless, to ensure transparency and reproducibility within these constraints, we provide access to the 5th-95th percentile confidence bands derived from the prior and posterior streamflow distributions. These are clearly referenced in the Data Availability section and enable readers to evaluate the uncertainty structure and relative discharge variability represented in the analysis. In addition, the Supplementary Material includes dimensionless time series and flow duration curves that were normalized using consistent scales across the three stations. This approach facilitates a fair comparison of streamflow magnitudes while preserving the relative differences between sites. In line with the rationale adopted by Hydrology and Earth System Sciences

regarding restricted datasets, these choices are clearly justified in the section on data availability and confidentiality.

.....

**Reply to Review 2:**

I acknowledge the efforts that the authors made in revising the manuscript and the point-to-point response to each comment. Many comments have been addressed, while a few comments are partially addressed. I still have some concerns about the results of the revised manuscript.

**Discharge:**

One of my key concerns is still that it is not possible to assess the simulated discharge performance. For example in Figure 5, the authors plot the discharge without y-axis. Only the best simulated discharge data of the posterior distributions are provided on the Zenodo (https://zenodo.org/records/15605202), however, they are not plotted on the Figure 5. The authors provided the simulation results of two more discharge stations in the supplementary material, but neither observed nor simulated data are provided for these stations. I would suggest the authors to keep consistency: the best simulation's data provided on the link should be plotted on the figures to allow readers to assess the differences between the observation, prior, and posterior distributions. The uncertainty bands are too wide to obtain this information and obscure the difference between the simulation and observation. The title of the manuscript is reducing hydrological streamflow uncertainty by using snow, glacier, and isotope data. However, the differences of the simulated discharge with these data are hardly seen in Figures 5, S6, S7 (a,c,e,b,d).

**Reply**

We thank the reviewer for their polite and constructive comment, and we are grateful for highlighting the importance of better representing the data used in our figures. This suggestion prompted us to further reflect on the role of ensemble means in the context of our modeling framework and to address additional important questions. In response, we have updated the Zenodo archive (https://zenodo.org/records/15605202) to include not only the 5-95% uncertainty bands but also the mean streamflow trajectories for both the prior and posterior ensembles at all three gauging stations (Nuxia, Yangcun, and Nugesha). These simulation means are the same as those now shown in Figure 5 of the main text and in Figures S6-S7 of the Supplementary Material.

We have also revised the manuscript accordingly. In the Results section, we added the following paragraph to enhance interpretability and provide a deterministic reference alongside the probabilistic representation:

"To further enhance interpretability and provide a deterministic reference alongside the probabilistic representation, Figure 5 includes the mean simulated streamflow trajectories for both the prior and posterior distributions, in addition to the uncertainty bands and observed data. As evident from the figure insets and the FDCs, the prior and posterior means exhibit slight differences across all cases, with a more noticeable divergence of the posterior mean from the prior in the case of isotope conditioning."

Likewise, in the Discussion section, we now explicitly discuss the role and limitations of the ensemble mean:

"In this context, the posterior mean streamflow, especially in the isotopeconditioned simulations, fails to consistently outperform the prior mean streamflow in reproducing the observed discharge, despite exhibiting narrower uncertainty bands in some streamflow regimes (see Section 3). This deterioration in deterministic skill is not unexpected. Previous studies (e.g., Vrugt and Sadegh, 2013; Botto et al., 2018) have shown that reducing ensemble spread does not automatically lead to improved agreement with observations. Structural model deficiencies and varying accuracy of input data sources (i.e., SCA, GMB, and I) may introduce systematic posterior bias, since the conditioning step attempts to compensate for processes that are poorly captured by the model or affected by different levels of uncertainty (Beven and Freer, 2001; Chowdhury and Sharma, 2007). It is important to emphasize that the ensemble mean does not correspond to the best-performing simulation in terms of NSE, and may smooth out dynamic features that are better reproduced by individual ensemble members. Moreover, the goal of the data-conditioning approach is not to maximize deterministic skill, but rather to reduce predictive uncertainty by constraining the prior ensemble: the shift from prior to posterior aims at narrowing the uncertainty bands of the streamflow simulations, even at the cost of some loss in individual accuracy (Beven, 2006)."

We believe that these additions directly address the reviewer's concern and strengthen the coherence between the figures, the shared data, and the overall objective of the study.

The simulated discharge for the other two stations (Fig. S6 and S7 only shown for 2005-2010) seem to be worse than Figure 5 (only shown for 2010-2015). The simulation period was set for 2001-2015. Can the authors please explain why the model is better for one station but worse for the other two stations? Why not show the overlapping period of all stations?

**Reply**

We thank the reviewer for this insightful observation. The lower performance at the two upstream stations is mainly due to the absence of site-specific calibration: parameter sets were calibrated at Nuxia and transferred unchanged to Yangcun and Nugesha, so they do not fully capture local hydrological behaviour. This outcome is consistent with earlier findings on parameter transferability (e.g., Khakbaz et al., 2012; Demirel et al., 2024). This clarification is now explicitly noted in Section 3.2 of the revised manuscript. We also acknowledge that, in the previous version, Figures 5, S6, and S7 did not cover the same time period at all stations, this was an oversight on our part. To ensure a fair and consistent comparison, we have redrawn these figures to span the same overlapping period (2001–2010) at all three sites. In addition, each panel now uses a uniform, dimensionless y-axis, which facilitates direct comparison and interpretation of differences across stations. To further support this comparison, we have added a new figure in the Supplementary Material (Figure S9), which directly contrasts the dimensionless observed streamflow time series across the three stations. This provides a clearer view of their relative hydrological regimes and supports the interpretation of ensemble performance discussed in the main text.

In Figure 5(b,d,f), S6(b,d,f), S7 (b,d,f), as there is no y-axis, I am not sure about the high and low discharge distribution, either upside or downside? I also do not understand the unevenly distributed ticks on y-axis.

**Reply**

We thank the reviewer for highlighting this issue. The unevenly spaced ticks in the original panels arose from plotting the flow-duration curves on a logarithmic axis. In the revised figures, the time-series panels now display normalized streamflow, while the FDC panels use a normalized log-discharge scale. This approach maintains the customary logarithmic representation of FDCs yet presents all values in a clear, dimensionless form improving overall readability.

Pareto-front: How do the authors define the Pareto fronts which are not dominated by both objectives in Figure 3 on L231?

**Reply**

We thank the reviewer for this helpful comment and acknowledge that our original phrasing was unclear. We have revised the relevant sentence to clarify how the Pareto front is defined in our analysis. Specifically, we follow standard practice in multi-objective hydrological modelling (Yapo et al., 1998, Efstratiadis and Koutsoyiannis, 2010), and define the Pareto front as the set of nondominated simulations; those for which no other simulation in the ensemble achieves equal or better performance in both objectives and strictly better in at least one. These points represent optimal trade-offs: improving one objective would necessarily deteriorate the other. We have updated the manuscript to reflect this more precise formulation.

**Tech corrections:**

In the data availability, the full names of the abbreviations should be given, e.g. CMFD, HWSD.

**Reply**

Thank you for your observation. We have updated the Data Availability section to include the full names of all abbreviations, including CMFD (China Meteorological Forcing Dataset) and HWSD (Harmonized World Soil Database), to improve clarity and ensure accessibility for all readers.

Figure 2: avoid using red and green color in the same figure to allow readers with color vision deficiency to correctly interpret the figure. This issue has been raised in last round of review but still has not been addressed.

**Reply**

We appreciate the reviewer's attention to accessibility and apologize for not having fully addressed this point in the previous revision. In the revised manuscript, we have updated Figure 2 to avoid the use of red and green in the same figure. The new color scheme has been carefully selected to be distinguishable for readers with color vision deficiency. Additionally, we have validated the updated figure using the Color Blindness Simulator available at https://www.color-blindness.com/coblis-color-blindness-simulator/ to ensure accessibility.

**Bibliography**

- Beven, K. (2006). A manifesto for the equifinality thesis. Journal of Hydrology, 320, 18–36. https://doi.org/10.1016/j.jhydrol.2005.07.007
- Beven, K., & Freer, J. (2001). Equifinality, data assimilation, and uncertainty estimation in mechanistic modelling of complex environmental systems using the GLUE methodology. Journal of Hydrology, 249, 11–29. https://doi.org/10.1016/S0022-1694(01)00421-8
- Botto, A., Belluco, E., & Camporese, M. (2018). Multi-source data assimilation for physically based hydrological modeling of an experimental hillslope. Hydrology and Earth System Sciences, 22, 4251–4266.

- https://doi.org/10.5194/hess-22-4251-2018
- Chowdhury, S., & Sharma, A. (2007). Mitigating parameter bias in hydrological modelling due to uncertainty in covariates. Journal of Hydrology, 340, 197–204. https://doi.org/10.1016/j.jhydrol.2007.04.010
- Efstratiadis, A., & Koutsoyiannis, D. (2010). One decade of multi-objective calibration approaches in hydrological modelling: A review. Hydrological Sciences Journal, 55, 58–78. https://doi.org/10.1080/02626660903526292
- Khakbaz, B., Imam, B., Hsu, K., & Sorooshian, S. (2012). From lumped to distributed via semi-distributed: Calibration strategies for semi-distributed hydrologic models. Journal of Hydrology, 418–419, 61–77. https://doi.org/10.1016/j.jhydrol.2009.02.021
- Lin, J., Bryan, B. A., Zhou, X., et al. (2023). Making China's water data accessible, usable and shareable. Nature Water, 1, 328–335. https://doi.org/10.1038/s44221-023-00039-y
- Singh, D., Vardhan, M., Sahu, R., Chatterjee, D., Chauhan, P., & Liu, S. (2023). Machine-learning- and deep-learning-based streamflow prediction in a hilly catchment for future scenarios using CMIP6 GCM data. Hydrology and Earth System Sciences, 27(5), 1047–1075. https://doi.org/10.5194/hess-27-1047-2023
- Vrugt, J. A., & Sadegh, M. (2013). Toward diagnostic model calibration and evaluation: Approximate Bayesian computation. Water Resources Research, 49, 4335–4345. https://doi.org/10.1002/wrcr.20354
- Yapo, P. O., Gupta, H. V., & Sorooshian, S. (1998). Multi-objective global optimization for hydrologic models. Journal of Hydrology, 204, 83–97. https://doi.org/10.1016/S0022-1694(97)00107-8
- Zhang, J., Solomatine, D., & Dong, Z. (2024). Robust multi-objective optimization under multiple uncertainties using the CM-ROPAR approach: Case study of water resources allocation in the Huaihe River basin. Hydrology and Earth System Sciences, 28(16), 3739–3753. https://doi.org/10.5194/hess-28-3739-2024